# Restructuring Vector Quantization with the Rotation Trick

**Christopher Fifty**[1]**, Ronald G. Junkins**[1]**, Dennis Duan**[1,2]**, Aniketh Iyengar**[1]**, Jerry W. Liu**[1]**,
Ehsan Amid**[2]**, Sebastian Thrun**[1]**, Christopher Ré**[1]
[1]Stanford University, [2]Google DeepMind
`fifty@cs.stanford.com`

## Abstract

Vector Quantized Variational AutoEncoders (VQ-VAEs) are designed to compress a continuous input to a discrete latent space and reconstruct it with minimal distortion. They operate by maintaining a set of vectors—often referred to as the codebook—and quantizing each encoder output to the nearest vector in the codebook. However, as vector quantization is non-differentiable, the gradient to the encoder flows *around* the vector quantization layer rather than *through* it in a straight-through approximation. This approximation may be undesirable as all information from the vector quantization operation is lost. In this work, we propose a way to propagate gradients through the vector quantization layer of VQ-VAEs. We smoothly transform each encoder output into its corresponding codebook vector via a rotation and rescaling linear transformation that is treated as a constant during backpropagation. As a result, the relative magnitude and angle between encoder output and codebook vector becomes encoded into the gradient as it propagates through the vector quantization layer and back to the encoder. Across 11 different VQ-VAE training paradigms, we find this restructuring improves reconstruction metrics, codebook utilization, and quantization error. Our code is available at https://github.com/cfifty/rotation_trick.

## 1 Introduction

Vector quantization (Gray, 1984) is an approach to discretize a continuous vector space. It defines a finite set of vectors—referred to as the codebook—and maps any vector in the continuous vector space to the closest vector in the codebook. However, deep learning paradigms that use vector quantization are often difficult to train because replacing a vector with its closest codebook counterpart is a non-differentiable operation (Huh et al., 2023). This characteristic was not an issue at its creation during the Renaissance of Information Theory for applications like noisy channel communication (Cover, 1999); however in the era deep learning, it presents a challenge as gradients cannot directly flow through layers that use vector quantization during backpropagation.

In deep learning, vector quantization is largely used in the eponymous Vector Quantized-Variational AutoEncoder (VQ-VAE) (Van Den Oord et al., 2017). A VQ-VAE is an AutoEncoder with a vector quantization layer between the encoder's output and decoder's input, thereby quantizing the learned representation at the bottleneck. While VQ-VAEs are ubiquitous in state-of-the-art generative modeling (Rombach et al., 2022; Dhariwal et al., 2020; Brooks et al., 2024), their gradients cannot flow from the decoder to the encoder uninterrupted as they must pass through a non-differentiable vector quantization layer.

A solution to the non-differentiability problem is to approximate gradients via a "straight-through estimator" (STE) (Bengio et al., 2013). During backpropagation, the STE copies and pastes the gradients from the decoder's input to the encoder's output, thereby skipping the quantization operation altogether. However, this approximation can lead to poor-performing models and codebook collapse: a phenomena where a large percentage of the codebook converge to zero norm and are unused by the model (Mentzer et al., 2023). Even if codebook collapse does not occur, the codebook is often under-utilized, thereby limiting the information capacity of the VQ-VAEs's bottleneck (Dhariwal et al., 2020).

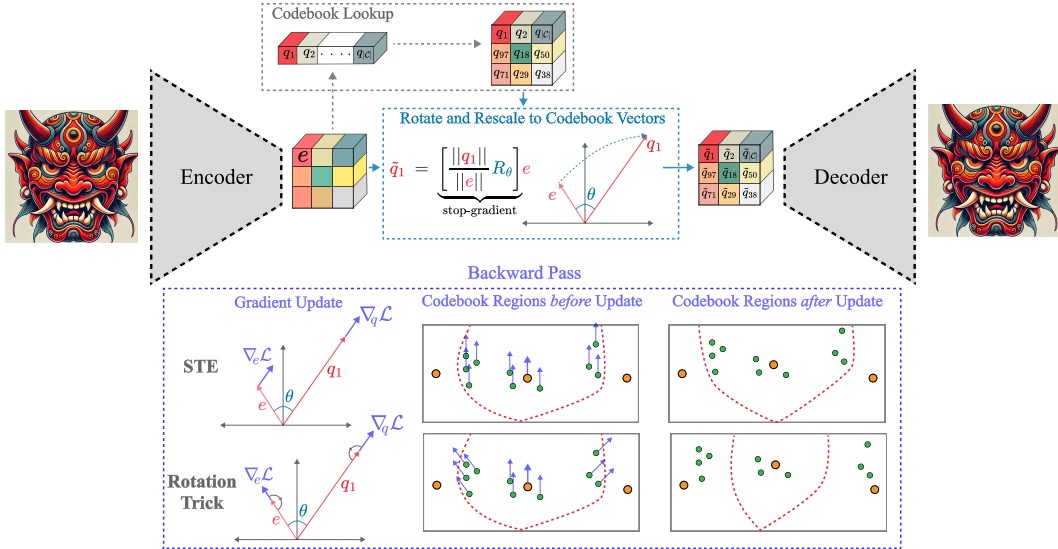

Figure 1: Illustration of the rotation trick. In the forward pass, encoder output $e$ is rotated and rescaled to $q_1$. For simplicity, the rotations of other encoder outputs are not shown. In the backward pass, the gradient at $q_1$ moves to $e$ so that the angle between $\nabla_{q_1}\mathcal{L}$ and $q_1$ is preserved. Now, points within the same codebook region receive different gradients depending on their relative angle and magnitude to the codebook vector. For example, points with high angular distance can be *pushed* into new codebook regions, thereby increasing codebook utilization.

In this work, we propose an alternate way to propagate gradients through the vector quantization layer in VQ-VAEs. For a given encoder output $e$ and nearest codebook vector $q$, we smoothly transform $e$ to $q$ via a rotation and rescaling linear transformation and then send this output—rather than the direct result of the codebook lookup—to the decoder. As the input to the decoder, $\tilde{q}$, is now treated as a smooth linear transformation of $e$, gradients flow back from the decoder to the encoder unimpeded. To avoid differentiating through the rotation and rescaling, we treat both as constants with respect to $e$ and $q$. We explain why

**Algorithm 1** The Rotation Trick

**Require:** input example $x$
  $e \leftarrow \text{Encoder}(x)$
  $q \leftarrow$ nearest codebook vector to $e$
  $R \leftarrow$ rotation matrix that aligns $e$ to $q$
  $\tilde{q} \leftarrow \text{stop-gradient}\left[\frac{\|q\|}{\|e\|}R\right]e$
  $\tilde{x} \leftarrow \text{Decoder}(\tilde{q})$
  $\text{loss} \leftarrow \mathcal{L}(x, \tilde{x})$
  **return** loss

this choice is necessary in Appendix A.7. Following the convention of Kingma & Welling (2013), we call this restructuring "the rotation trick." It is illustrated in Figure 1 and described in Algorithm 1.

The rotation trick does not change the output of the VQ-VAE in the forward pass. However, during the backward pass, it transports the gradient $\nabla_q\mathcal{L}$ at $q$ to become the gradient $\nabla_e\mathcal{L}$ at $e$ so that the angle between $q$ and $\nabla_q\mathcal{L}$ *after* the vector quantization layer equals the angle between $e$ and $\nabla_e\mathcal{L}$ *before* the vector quantization layer. Preserving this angle encodes relative angular distances and magnitudes into the gradient and changes how points within the same codebook region are updated.

The STE applies the same update to all points within the same codebook region, maintaining their relative distances. However as we will show in Section 4.3, the rotation trick can push points within the same codebook region farther apart—or pull them closer together—depending on the direction of the gradient vector. The former capability can correspond to increased codebook usage while the latter to lower quantization error. In the context of lossy compression, both capabilities are desirable for reducing the distortion and increasing the information capacity of the vector quantization layer.

When applied to several open-source VQ-VAE repositories, we find the rotation trick substantively improves reconstruction performance, increases codebook usage, and decreases the distance between encoder outputs and their corresponding codebook vectors. For instance, training the VQGAN from Rombach et al. (2022) on ImageNet (Deng et al., 2009) with the rotation trick improves reconstruction FID from 5.0 to 1.1, reconstruction IS from 141.5 to 200.2, increases codebook usage from 2% to 27%, and decreases quantization error by two orders of magnitude.

## 2 RELATED WORK

Many researchers have built upon the seminal work of Van Den Oord et al. (2017) to improve VQ-VAE performance. While non-exhaustive, our review focuses on methods that address training

instabilities caused by the vector quantization layer. We partition these efforts into two categories: (1) methods that sidestep the STE and (2) methods that improve codebook-model interactions.

**Sidestepping the STE.** Several prior works have sought to fix the problems caused by the STE by avoiding deterministic vector quantization. Baevski et al. (2019) employ the Gumbel-Softmax trick (Jang et al., 2016) to fit a categorical distribution over codebook vectors that converges to a one-hot distribution towards the end of training, Gautam et al. (2023) quantize using a convex combination of codebook vectors, and Takida et al. (2022) employ stochastic quantization. Unlike the above that cast vector quantization as a distribution over codebook vectors, Huh et al. (2023) propose an alternating optimization where the encoder is optimized to output representations close to the codebook vectors while the decoder minimizes reconstruction loss from a fixed set of codebook vector inputs. While these approaches sidestep the training instabilities caused by the STE, they can introduce their own set of problems and complexities such as low codebook utilization at inference and the tuning of a temperature schedule (Zhang et al., 2023). As a result, many applications and research papers continue to employ VQ-VAEs that are trained using the STE (Rombach et al., 2022; Chang et al., 2022; Huang et al., 2023; Zhu et al., 2023; Dong et al., 2023).

**Codebook-Model Improvements.** Another way to attack codebook collapse or under-utilization is to change the codebook lookup. Rather than use Euclidean distance, Yu et al. (2021) employ a cosine similarity measure, Goswami et al. (2024) a hyperbolic metric, and Lee et al. (2022) stochastically sample codes as a function of the distance between the encoder output and codebook vectors. Another perspective examines the learning of the codebook. Kolesnikov et al. (2022) split high-usage codebook vectors, Dhariwal et al. (2020); Łańcucki et al. (2020); Zheng & Vedaldi (2023) resurrect low-usage codebook vectors throughout training, Chen et al. (2024) dynamically selects one of $m$ codebooks for each datapoint, and Mentzer et al. (2023); Zhao et al. (2024); Yu et al. (2023); Chiu et al. (2022) fix the codebook vectors to an *a priori* geometry and train the model without learning the codebook at all. Other works propose loss penalties to encourage codebook utilization. Zhang et al. (2023) add a KL-divergence penalty between codebook utilization and a uniform distribution while Yu et al. (2023) add an entropy loss term to penalize low codebook utilization. While effective at targeting specific training difficulties, as each of these methods continue to use the STE, the training instability caused by this estimator persist. Most of our experiments in Section 5 implement a subset of these approaches, and we find that replacing the STE with the rotation trick further improves performance.

## 3 STRAIGHT THROUGH ESTIMATOR (STE)

In this section, we review the Straight-Through Estimator (STE) and visualize its effect on the gradients. We then explore two STE alternatives that—at first glance—appear to correct the approximation made by the STE.

For notation, we define a sample space $\mathcal{X}$ over the input data with probability distribution $p$. For input $x \in \mathcal{X}$, we define the encoder as a deterministic mapping that parameterizes a posterior distribution $p_{\mathcal{E}}(e|x)$. The vector quantization layer, $\mathcal{Q}(\cdot)$, is a function that selects the codebook vector $q \in \mathcal{C}$ nearest to the encoder output $e$. Under Euclidean distance, it has the form:

$$\mathcal{Q}(q = i|e) = \begin{cases} 1 \text{ if } i = \arg\min_{1 \leq j \leq |\mathcal{C}|} \|e - q_j\|_2 \\ 0 \text{ otherwise} \end{cases}$$

The decoder is similarly defined as a deterministic mapping that parameterizes the conditional distribution over reconstructions $p_{\mathcal{D}}(\tilde{x}|q)$. As in the VAE (Kingma & Welling, 2013), the loss function follows from the ELBO with the KL-divergence term zeroing out as $p_{\mathcal{E}}(e|x)$ is deterministic and the utilization over codebook vectors is assumed to be uniform. Van Den Oord et al. (2017) additionally add a "codebook loss" term $\|sg(e) - q\|_2^2$ to learn the codebook vectors and a "commitment loss" term $\beta\|e - sg(q)\|_2^2$ to pull the encoder's output towards the codebook vectors. $sg$ stands for stop-gradient and $\beta$ is a hyperparameter, typically set to a value in $[0.25, 2]$. For predicted reconstruction $\tilde{x}$, the optimization objective becomes:

$$\mathcal{L}(\tilde{x}) = \|x - \tilde{x}\|_2^2 + \|sg(e) - q\|_2^2 + \beta\|e - sg(q)\|_2^2$$

In the subsequent analysis, we focus only on the $\|x - \tilde{x}\|_2^2$ term as the other two are not functions of the decoder. During backpropagation, the model must differentiate through the vector quantization

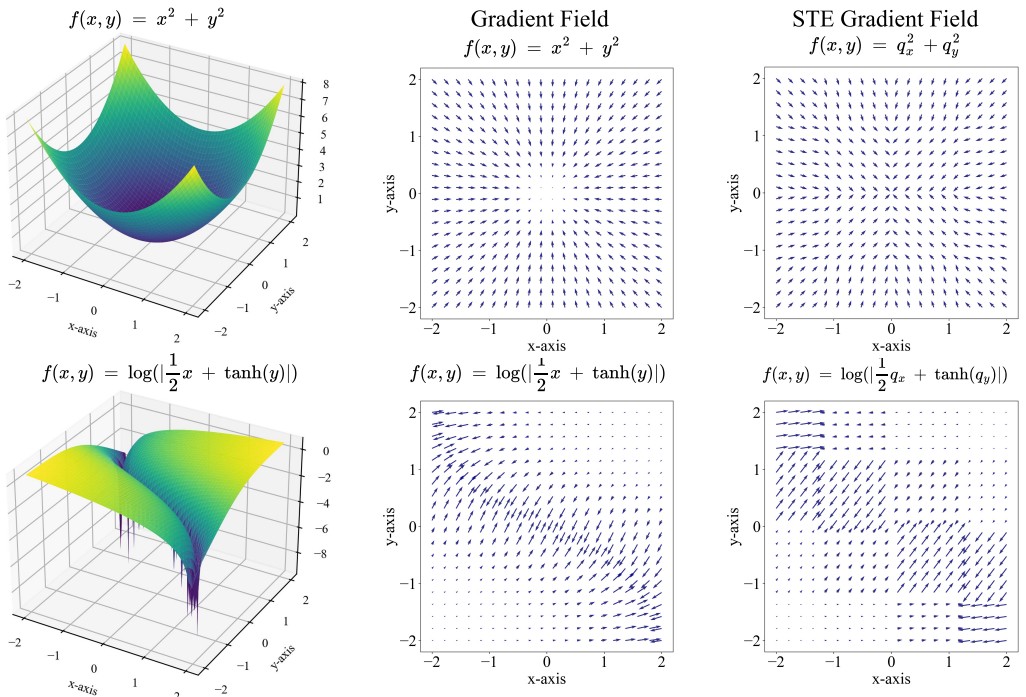

Figure 2: Visualization of how the straight-through estimator (STE) transforms the gradient field for 16 codebook vectors for (top) $f(x, y) = x^2 + y^2$ and (bottom) $f(x, y) = \log\left(|\frac{1}{2}x + \tanh(y)|\right)$. The STE takes the gradient at the codebook vector $(q_x, q_y)$ and "copies-and-pastes" it to all other locations within the same codebook region, forming a "checker-board" pattern in the gradient field.

function $\mathcal{Q}(\cdot)$. We can break down the backward pass into three terms:

$$\frac{\partial \mathcal{L}}{\partial x} = \frac{\partial \mathcal{L}}{\partial q} \frac{\partial q}{\partial e} \frac{\partial e}{\partial x}$$

where $\frac{\partial \mathcal{L}}{\partial q}$ represents backpropagation through the decoder, $\frac{\partial q}{\partial e}$ represents backpropagation through the vector quantization layer, and $\frac{\partial e}{\partial x}$ represents backpropagation through the encoder. As vector quantization is not a smooth transformation, $\frac{\partial q}{\partial e}$ cannot be computed and gradients cannot flow through this term to update the encoder in backpropagation.

To solve the issue of non-differentiability, the STE copies the gradients from $q$ to $e$, bypassing vector quantization entirely. Simply, the STE sets $\frac{\partial q}{\partial e}$ to the identity matrix $I$ in the backward pass:

$$\frac{\partial \mathcal{L}}{\partial x} = \frac{\partial \mathcal{L}}{\partial q} I \frac{\partial e}{\partial x}$$

The first two terms $\frac{\partial \mathcal{L}}{\partial q} \frac{\partial q}{\partial e}$ combine to $\frac{\partial \mathcal{L}}{\partial e}$ which, somewhat misleadingly, does not actually depend on $e$. As a consequence, the location of $e$ within the Voronoi partition generated by codebook vector $q$—be it close to $q$ or at the boundary of the region—has no impact on the gradient update to the encoder.

An example of this effect is visualized in Figure 2 for two example functions. In the STE approximation, the "exact" gradient at the encoder output is replaced by the gradient at the corresponding codebook vector for each Voronoi partition, irrespective of where in that region the encoder output $e$ lies. As a result, the exact gradient field becomes "partitioned" into 16 different regions—all with the same gradient update to the encoder—for the 16 vectors in the codebook.

Returning to our question, is there a better way to propagate gradients through the vector quantization layer? At first glance, one may be tempted to estimate the curvature at $q$ and use this information to transform $\frac{\partial q}{\partial e}$ as $q$ moves to $e$. This is accomplished by taking a second order expansion around $q$ to approximate the value of the loss at $e$:

$$\mathcal{L}_e \approx \mathcal{L}_q + (\nabla_q \mathcal{L})^T (e - q) + \frac{1}{2}(e - q)^T (\nabla_q^2 \mathcal{L})(e - q)$$

Then we can compute the gradient at the point $e$ instead of $q$ up to second order approximation with:

$$\frac{\partial \mathcal{L}}{\partial e} \approx \frac{\partial}{\partial e} \left[ \mathcal{L}_q + (\nabla_q \mathcal{L})^T (e - q) + \frac{1}{2}(e - q)^T (\nabla_q^2 \mathcal{L})(e - q) \right]$$
$$= \nabla_q \mathcal{L} + (\nabla_q^2 \mathcal{L})(e - q)$$

While computing Hessians with respect to model parameters are typically prohibitive in modern deep learning architectures, computing them with respect to only the codebook is feasible. Moreover as we must only compute $(\nabla_q^2 \mathcal{L})(e - q)$, one may take advantage of efficient Hessian-Vector products implementations in deep learning frameworks (Dagréou et al., 2024) and avoid computing the full Hessian matrix.

Extending this idea a step further, we can compute the exact gradient $\frac{\partial \mathcal{L}}{\partial e}$ at $e$ by making two passes through the network. Let $\mathcal{L}_q$ be the loss with the vector quantization layer and $\mathcal{L}_e$ be the loss without vector quantization, i.e. $q = e$ rather than $q = \mathcal{Q}(e)$. Then one may form the total loss $\mathcal{L} = \mathcal{L}_q + \lambda \mathcal{L}_e$, where $\lambda$ is a small constant like $10^{-6}$, to scale down the effect of $\mathcal{L}_e$ on the decoder's parameters and use a gradient scaling multiplier of $\lambda^{-1}$ to reweigh the effect of $\mathcal{L}_e$ on the encoder's parameters to 1. As $\frac{\partial q}{\partial e}$ is non-differentiable, gradients from $\mathcal{L}_q$ will not flow to the encoder.

**While seeming to correct the encoder's gradients, replacing the STE with either approach will likely result in worse performance.** This is because computing the exact gradient with respect to $e$ is actually the AutoEncoder (Hinton & Zemel, 1993) gradient, the model that VAEs (Kingma & Welling, 2013) and VQ-VAEs (Van Den Oord et al., 2017) were designed to replace given the AutoEncoder's propensity to overfit and difficultly generalizing. Accordingly using either Hessian approximation or exact gradients via a double forward pass will cause the encoder to be trained like an AutoEncoder and the decoder to be trained like a VQ-VAE. This mis-match in optimization objectives is likely another contributing factor to the poor performance we observe for both methods in Table 1, and a deeper analysis into these characteristics is presented in Appendix A.3.

## 4 THE ROTATION TRICK

As discussed in Section 3, updating the encoder's parameters by approximating, or exactly, computing the gradient at the encoder's output is undesirable. Similarly, the STE appears to lose information: the location of $e$ within the quantized region—be it close to $q$ or far away at the boundary—has no impact on the gradient update to the encoder. Capturing this information, i.e. using the location of $e$ in relation to $q$ to transform the gradients through $\frac{\partial q}{\partial e}$, could be beneficial to the encoder's gradient updates and an improvement over the STE.

Viewed geometrically, we ask how to move the gradient $\nabla_q \mathcal{L}$ from $q$ to $e$, and what characteristics of $\nabla_q \mathcal{L}$ and $q$ should be preserved during this movement. The STE offers one possible answer: move the gradient from $q$ to $e$ so that its direction and magnitude are preserved. However, this paper supplies a different answer: move the gradient so that the angle between $\nabla_q \mathcal{L}$ and $q$ is preserved as $\nabla_q \mathcal{L}$ moves to $e$. We term this approach "the rotation trick", and in Section 4.3 we show that preserving the angle between $q$ and $\nabla_q \mathcal{L}$ conveys desirable properties to how points move within the same quantized region.

### 4.1 THE ROTATION TRICK PRESERVES ANGLES

In this section, we formally define the rotation trick. For encoder output $e$, let $q = \mathcal{Q}(e)$ represent the corresponding codebook vector. $\mathcal{Q}(\cdot)$ is non-differentiable so gradients cannot flow through this layer during the backward pass. The STE solves this problem—maintaining the direction and magnitude of the gradient $\nabla_q \mathcal{L}$—as $\nabla_q \mathcal{L}$ moves from $q$ to $e$ with some clever hacking of the backpropagation function in deep learning frameworks:

$$\tilde{q} = e - \underbrace{(q - e)}_{\text{constant}}$$

which is a parameterization of vector quantization that sets the gradient at the encoder output to the gradient at the decoder's input. The rotation trick offers a different parameterization: casting the forward pass as a rotation and rescaling that aligns $e$ with $q$:

$$\tilde{q} = \underbrace{\left[ \frac{\|q\|}{\|e\|} R \right]}_{\text{constant}} e$$

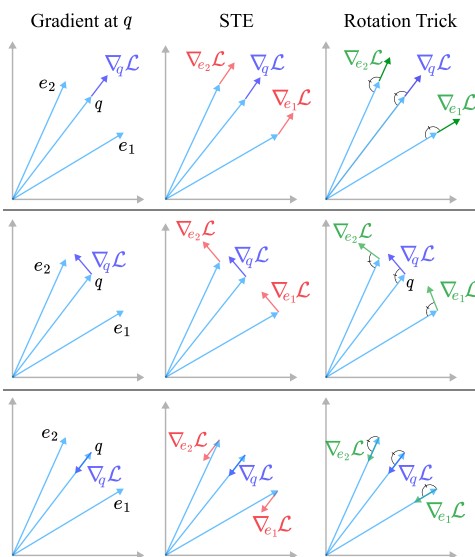

$R$ is the rotation[1] transformation that aligns $e$ with $q$ and $\frac{\|q\|}{\|e\|}$ rescales $e$ to have the same magnitude as $q$. Note that both $R$ and $\frac{\|q\|}{\|e\|}$ are functions of $e$. To avoid differentiating through this dependency, we treat them as fixed constants—or detached from the computational graph in deep learning frameworks—when differentiating. This choice is explained in Appendix A.7.

While the rotation trick does not change the output of the forward pass, the backward pass changes. Rather than set $\frac{\partial q}{\partial e} = I$ as in the STE, the rotation trick sets $\frac{\partial q}{\partial e}$ to be a rotation and rescaling transformation:

$$\frac{\partial \tilde{q}}{\partial e} = \frac{\|q\|}{\|e\|} R$$

As a result, $\frac{\partial q}{\partial e}$ changes based on the position of $e$ in the codebook partition of $q$, and notably, the angle

Figure 3: Illustration of how the gradient at $q$ moves to $e$ via the STE (middle) and rotation trick (right). The STE "copies-and-pastes" the gradient to preserve its direction while the rotation trick moves the gradient so the angle between $q$ and $\nabla_q \mathcal{L}$ is preserved (proved in Appendix A.6).

between $\nabla_q \mathcal{L}$ and $q$ is preserved as $\nabla_q \mathcal{L}$ moves to $e$. This effect is visualized in Figure 3. While the STE translates the gradient from $q$ to $e$, the rotation trick rotates it so that the angle between $\nabla_q \mathcal{L}$ and $q$ is preserved. In a sense, the rotation trick and the STE are siblings. They choose different characteristics of the gradient as desiderata and then preserve those characteristics as the gradient flows around the non-differentiable vector quantization operation to the encoder.

### 4.2 EFFICIENT ROTATION COMPUTATION

The rotation transformation $R$ that rotates $e$ to $q$ can be efficiently computed with Householder matrix reflections. We define $\hat{e} = \frac{e}{\|e\|}$, $\hat{q} = \frac{q}{\|q\|}$, $\lambda = \frac{\|q\|}{\|e\|}$, and $r = \frac{\hat{e}+\hat{q}}{\|\hat{e}+\hat{q}\|}$. Then the rotation and rescaling that aligns $e$ to $q$ is simply:

$$\begin{aligned}
\tilde{q} &= \lambda R e \\
&= \lambda (I - 2rr^T + 2\hat{q}\hat{e}^T) e \\
&= \lambda [e - 2rr^T e + 2\hat{q}\hat{e}^T e]
\end{aligned}$$

Due to space constraints, we leave the derivation of this formula to Appendix A.5. Parameterizing the rotation in this fashion avoids computing outer products and therefore consumes minimal GPU VRAM. Further, we did not detect a difference in wall-clock time between VQ-VAEs trained with the STE and VQ-VAEs trained with the rotation trick for our experiments in Section 5.

### 4.3 VORONOI PARTITION ANALYSIS

In the context of lossy compression, vector quantization works well when the distortion, or equivalently quantization error $\|e - q\|_2^2$, is low and the information capacity—equivalently codebook utilization—is high (Cover, 1999). Later in Section 5, we will see that VQ-VAEs trained with the rotation trick have this *desiderata*—often reducing quantization error by an order of magnitude and substantially increasing codebook usage—when compared to VQ-VAEs trained with the STE. However, the underlying reason *why* this occurs is less clear.

---

[1] A rotation is defined as a linear transformation so that $RR^T = I$, $R^{-1} = R^T$, and $\det(R) = 1$.

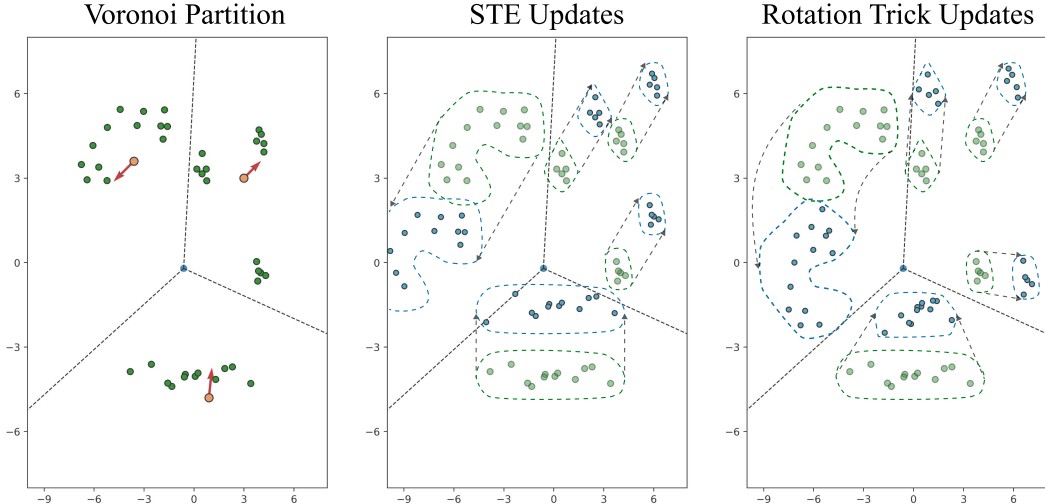

Figure 4: Depiction of how points within the same codebook region change after a gradient update (red arrow) at the codebook vector (orange circle). The STE applies the same update to each point in the same region. The rotation trick modifies the update based on the location of each point with respect to the codebook vector.

In this section, we analyze the effect of the rotation trick by looking at how encoder outputs that are mapped to the same Voronoi region are updated. While the STE applies the same update to all points within the same partition, the rotation trick changes the update based on the location of points within the Voronoi region. It can push points within the same region farther apart or pull them closer together depending on the direction of the gradient vector. The former capability can correspond to increased codebook usage while the latter to lower quantization error.

Let $\theta$ be the angle between $e$ and $q$ and $\phi$ be the angle between $q$ and $\nabla_q \mathcal{L}$. When $\nabla_q \mathcal{L}$ and $q$ point in the same direction, i.e. $-\pi/2 < \phi < \pi/2$, encoder outputs with large angular distance to $q$ are pushed *farther* away than they would otherwise be moved by the STE update. Figure 5 illustrates this effect. The points with large angular distance (blue regions) move further away from $q$ than the points with low angular distance (ivory regions).

The top right partitions of Figure 4 present an example of this effect. The two clusters of points at the boundary—with relatively large angle to the codebook vector—are pushed away while the cluster of points with small angle to the codebook vector move with it. The ability to push points at the boundary out of a quantized region and into another is desirable for increasing codebook utilization. Specifically, codebook utilization improves when points are pushed into the Voronoi regions of previously unused codebook vectors. This capability is not shared by the STE, which moves all points in the same region by the same amount.

When $\nabla_q \mathcal{L}$ and $q$ point in opposite directions, i.e. $\pi/2 < \phi < 3\pi/2$, the distance among points within the same Voronoi region decreases as they are pulled towards the location of the updated codebook vector. This effect is visualized in Figure 5 (green regions) and the bottom partitions of Figure 4 show an example. Unlike the STE update—that maintains the distances among points—the rotation trick pulls points with high angular distances closer towards the post-update codebook vector. This capability is desirable for reducing the quantization error and enabling the encoder to *lock on* (Van Den Oord et al., 2017) to a target codebook vector.

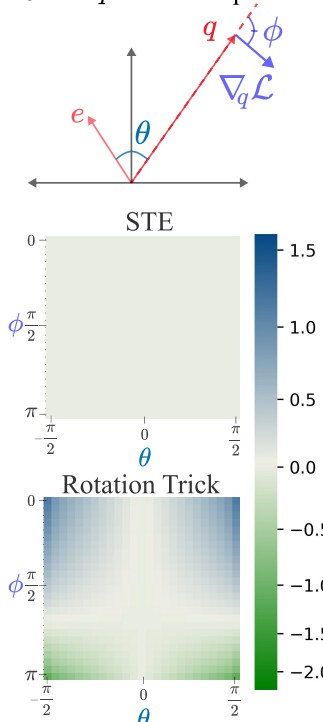

Figure 5: With the STE, the distances among points within the same region do not change. However with the rotation trick, the distances among points *do* change. When $\phi < \pi/2$, points with large angular distance are pushed away (blue: increasing distance). When $\phi > \pi/2$, points are *pulled* towards the codebook vector (green: decreasing distance).

Taken together, both capabilities can form a push-pull effect that achieves two *desiderata* of vector quantization: increasing information capacity and reducing distortion. Encoder outputs that have large

Table 1: Comparison of VQ-VAEs trained on ImageNet following Van Den Oord et al. (2017). We use the Vector Quantization layer from https://github.com/lucidrains/vector-quantize-pytorch.

| Approach | Training Metrics | | | Validation Metrics | | |
|---|---|---|---|---|---|---|
| | Codebook Usage ($\uparrow$) | Rec. Loss ($\downarrow$) | Quantization Error ($\downarrow$) | Rec. Loss ($\downarrow$) | r-FID ($\downarrow$) | r-IS ($\uparrow$) |
| Codebook Lookup: Euclidean & Latent Shape: $32 \times 32 \times 32$ & Codebook Size: 1024 | | | | | | |
| VQ-VAE | 100% | 0.107 | 5.9e-3 | **0.115** | 106.1 | 11.7 |
| VQ-VAE w/ Rotation Trick | 97% | 0.116 | 5.1e-4 | 0.122 | **85.7** | **17.0** |
| Codebook Lookup: Cosine & Latent Shape: $32 \times 32 \times 32$ & Codebook Size: 1024 | | | | | | |
| VQ-VAE | 75% | 0.107 | 2.9e-3 | 0.114 | 84.3 | 17.7 |
| VQ-VAE w/ Rotation Trick | 91% | 0.105 | 2.7e-3 | **0.111** | **82.9** | **18.1** |
| Codebook Lookup: Euclidean & Latent Shape: $64 \times 64 \times 3$ & Codebook Size: 8192 | | | | | | |
| VQ-VAE | 100% | 0.028 | 1.0e-3 | **0.030** | 19.0 | 97.3 |
| Gumbel VQ-VAE | 39% | 0.054 | — | 0.058 | 28.6 | 74.9 |
| VQ-VAE w/ Hessian Approx. | 39% | 0.082 | 6.9e-5 | 0.112 | 35.6 | 65.1 |
| VQ-VAE w/ Exact Gradients | 84% | 0.050 | 2.0e-3 | 0.053 | 25.4 | 80.4 |
| VQ-VAE w/ Rotation Trick | 99% | 0.028 | 1.4e-4 | **0.030** | **16.5** | **106.3** |
| Codebook Lookup: Cosine & Latent Shape: $64 \times 64 \times 3$ & Codebook Size: 8192 | | | | | | |
| VQ-VAE | 31% | 0.034 | 1.2e-4 | 0.038 | 26.0 | 77.8 |
| VQ-VAE w/ Hessian Approx. | 37% | 0.035 | 3.8e-5 | 0.037 | 29.0 | 71.5 |
| VQ-VAE w/ Exact Gradients | 38% | 0.035 | 3.6e-5 | 0.037 | 28.2 | 75.0 |
| VQ-VAE w/ Rotation Trick | 38% | 0.033 | 9.6e-5 | **0.035** | **24.2** | **83.9** |

angular distance to the chosen codebook vector are "pushed" to other, possibly unused, codebook regions by outwards-pointing gradients, thereby increasing codebook utilization. Concurrent with this effect, center-pointing gradients will "pull" points loosely clustered around the codebook vector closer together, locking on to the chosen codebook vector and reducing quantization error.

### 4.4 FURTHER ANALYSIS

The Appendix contains several supplementary analyses. Appendix A.2 compares the rotation trick with the STE for a non-convex synthetic example; Appendix A.4 looks at the behavior far away from the origin; and Appendix A.8 analyzes the effect of using a reflection rather than a rotation. Finally, Appendix A.9 examines scaling the gradient's norm by $\frac{\|q\|}{\|e\|}$ and explores alternatives.

## 5 EXPERIMENTS

In Section 4.3, we showed the rotation trick enables behavior that would increase codebook utilization and reduce quantization error by changing how points within the same Voronoi region are updated. However, the extent to which these changes will affect applications is unclear. In this section, we evaluate the effect of the rotation trick across many different VQ-VAE paradigms.

We begin with image reconstruction: training a VQ-VAE with the reconstruction objective of Van Den Oord et al. (2017) and later extend our evaluation to the more complex VQGANs (Esser et al., 2021), the VQGANs designed for latent diffusion (Rombach et al., 2022), and then the ViT-VQGAN (Yu et al., 2021). Finally, we evaluate VQ-VAE reconstructions on videos using a TimeSformer (Bertasius et al., 2021) encoder and decoder. Due to space constraints, the video results are presented in Appendix A.1. In total, our empirical analysis spans 11 different VQ-VAE configurations. For all experiments, aside from handling $\frac{\partial q}{\partial e}$ differently, the models, hyperparameters, and training settings are identical and described in Appendix A.10.

### 5.1 VQ-VAE EVALUATION

We begin with a straightforward evaluation: training a VQ-VAE to reconstruct examples from ImageNet (Deng et al., 2009). Following Van Den Oord et al. (2017), our training objective is a linear combination of the reconstruction, codebook, and commitment loss:

$$\mathcal{L} = \|x - \tilde{x}\|_2^2 + \|sg(e) - q\|_2^2 + \beta\|e - sg(q)\|_2^2$$

where $\beta$ is a hyperparameter scaling constant. Following convention, we drop the codebook loss term from the objective and instead use an exponential moving average to update the codebook vectors.

**Evaluation Settings.** For $256 \times 256 \times 3$ input images, we evaluate two different settings: (1) compressing to a latent space of dimension $32 \times 32 \times 32$ with a codebook size of 1024 following Yu et al. (2021) and (2) compressing to $64 \times 64 \times 3$ with a codebook size of 8192 following Rombach et al. (2022). In both settings, we compare with a Euclidean and cosine similarity codebook lookup.

Table 2: Results for VQGAN designed for autoregressive generation as implemented in https://github.com/CompVis/taming-transformers. Experiments on ImageNet and the combined dataset FFHQ (Karras et al., 2019) and CelebA-HQ (Karras, 2017) use a latent bottleneck of dimension $16 \times 16 \times 256$ with 1024 codebook vectors.

| Approach | Dataset | Codebook Usage | Quantization Error ($\downarrow$) | Valid Loss ($\downarrow$) | r-FID ($\downarrow$) | r-IS ($\uparrow$) |
|---|---|---|---|---|---|---|
| VQGAN (reported) | ImageNet | — | — | — | 7.9 | 114.4 |
| VQGAN (our run) | ImageNet | 95% | 0.134 | 0.594 | 7.3 | 118.2 |
| VQGAN w/ Rotation Trick | ImageNet | 98% | 0.002 | **0.422** | **4.6** | **146.5** |
| VQGAN | FFHQ & CelebA-HQ | 27% | 0.233 | 0.565 | 4.7 | 5.0 |
| VQGAN w/ Rotation Trick | FFHQ & CelebA-HQ | 99% | 0.002 | **0.313** | **3.7** | **5.2** |

Table 3: Results for VQGAN designed for latent diffusion as implemented in https://github.com/CompVis/latent-diffusion. Both settings train on ImageNet.

| Approach | Latent Shape | Codebook Size | Codebook Usage | Quantization Error ($\downarrow$) | Valid Loss ($\downarrow$) | r-FID ($\downarrow$) | r-IS ($\uparrow$) |
|---|---|---|---|---|---|---|---|
| VQGAN | $64 \times 64 \times 3$ | 8192 | 15% | 2.5e-3 | 0.183 | 0.53 | 220.6 |
| Gumbel VQGAN | $64 \times 64 \times 3$ | 8192 | 4% | — | 0.197 | 0.60 | 219.7 |
| VQGAN w/ Rotation Trick | $64 \times 64 \times 3$ | 8192 | 86% | 1.7e-4 | **0.142** | **0.27** | **228.0** |
| VQGAN | $32 \times 32 \times 4$ | 16384 | 2% | 1.2e-2 | 0.385 | 5.0 | 141.5 |
| Gumbel VQGAN | $32 \times 32 \times 4$ | 16384 | 12% | — | 0.3031 | 1.7 | 189.5 |
| VQGAN w/ Rotation Trick | $32 \times 32 \times 4$ | 16384 | 27% | 2.4e-4 | **0.269** | **1.1** | **200.2** |

**Evaluation Metrics.** We log both training and validation set reconstruction metrics. Of note, we compute reconstruction FID (Heusel et al., 2017) and reconstruction IS (Salimans et al., 2016) on reconstructions from the full ImageNet validation set as a measure of reconstruction quality. We also compute codebook usage, or the percentage of codebook vectors that are used in each batch of data, as a measure of the information capacity of the vector quantization layer and quantization error $\|e - q\|_2^2$ as a measure of distortion.

**Baselines.** Our comparison spans the STE estimator (*VQ-VAE*), stochastic quantization with Gumbel-Softmax (Baevski et al., 2019), (*Gumbel VQ-VAE*) the Hessian approximation described in Section 3 (*VQ-VAE w/ Hessian Approx*), the exact gradient backward pass described in Section 3 (*VQ-VAE w/ Exact Gradients*), and the rotation trick (*VQ-VAE w/ Rotation Trick*). All methods share the same architecture, hyperparameters, and training settings, and these settings are summarized in Table 8 of the Appendix. There is no functional difference among methods in the forward pass; the only differences relates to how gradients are propagated through $\frac{\partial q}{\partial e}$ during backpropagation.

**Results.** Table 1 displays our findings. We find that using the rotation trick reduces the quantization error—sometimes by an order of magnitude—and improves low codebook utilization. Both results are expected given the Voronoi partition analysis in Section 4.3: points at the boundary of quantized regions are likely pushed to under-utilized codebook vectors while points loosely grouped around the codebook vector are condensed towards it. These two features appear to have a meaningful effect on reconstruction metrics: training a VQ-VAE with the rotation trick substantially improves r-FID and r-IS.

We also see that the Hessian Approximation or using Exact Gradients results in poor reconstruction performance. While the gradients to the encoder are, in a sense, "more accurate", training the encoder like an AutoEncoder (Hinton & Zemel, 1993) likely introduces overfitting and poor generalization. Moreover, the mismatch in training objectives between the encoder and decoder is likely an aggravating factor and partly responsible for both models' poor performance.

## 5.2 VQGAN EVALUATION

Moving to the next level of complexity, we evaluate the effect of the rotation trick on VQGANs (Esser et al., 2021). The VQGAN training objective is:

$$\mathcal{L}_{\text{VQGAN}} = \mathcal{L}_{\text{Per}} + \|sg(e) - q\|_2^2 + \beta\|e - sg(q)\|_2^2 + \lambda\mathcal{L}_{\text{Adv}}$$

where $\mathcal{L}_{\text{Per}}$ is the perceptual loss from Johnson et al. (2016) and replaces the $L_2$ loss used to train VQ-VAEs. $L_{Adv}$ is a patch-based adversarial loss similar to the adversarial loss in Conditional GAN (Isola et al., 2017). $\beta$ is a constant that weights the commitment loss while $\lambda$ is an adaptive weight based on the ratio of $\nabla\mathcal{L}_{\text{Per}}$ to $\nabla\mathcal{L}_{\text{Adv}}$ with respect to the last layer of the decoder.

**Experimental Settings.** We evaluate VQGANs under two settings: (1) the paradigm amenable to autoregressive modeling with Transformers as described in Esser et al. (2021) and (2) the paradigm suitable to latent diffusion models as described in Rombach et al. (2022). The first setting follows the convolutional neural network and default hyperparameters described in Esser et al. (2021) while

Table 4: Results for ViT-VQGAN (Yu et al., 2021) trained on ImageNet. The latent shape is $8 \times 8 \times 32$ with 8192 codebook vectors. r-FID and r-IS are reported on the validation set.

| Approach | Codebook Usage (↑) | Train Loss (↓) | Quantization Error (↓) | Valid Loss (↓) | r-FID (↓) | r-IS (↑) |
|---|---|---|---|---|---|---|
| ViT-VQGAN [reported] | — | — | — | — | 22.8 | 72.9 |
| ViT-VQGAN [ours] | 0.3% | 0.124 | 6.7e-3 | 0.127 | 29.2 | 43.0 |
| ViT-VQGAN w/ Rotation Trick | 2.2% | 0.113 | 8.3e-3 | **0.113** | **11.2** | **93.1** |

the second follows those from Rombach et al. (2022). A full description of both training settings is provided in Table 9 of the Appendix.

**Results.** Our results are listed in Table 2 for the first setting and Table 3 for the second. Similar to our findings in Section 5.1, we find that training a VQ-VAE with the rotation trick substantially decreases quantization error and improves codebook usage. Moreover, reconstruction performance as measured on the validation set by the total loss, r-FID, and r-IS are improved across both modeling paradigms.

### 5.3 ViT-VQGAN Evaluation

Improving upon the VQGAN model, Yu et al. (2021) propose using a ViT (Dosovitskiy, 2020) rather than CNN to parameterize the encoder and decoder. The ViT-VQGAN uses factorized codes and $L_2$ normalization on the output and input to the vector quantization layer to improve performance and training stability. Additionally, the authors change the training objective, adding a logit-laplace loss and restoring the $L_2$ reconstruction error to $\mathcal{L}_{\text{VQGAN}}$.

**Experimental Settings.** We follow the open source implementation of https://github.com/thuanz123/enhancing-transformers and use the default model and hyperparameter settings for the small ViT-VQGAN. A complete description of the training settings can be found in Table 10 of the Appendix.

**Results.** Table 4 summarizes our findings. Similar to our previous results for VQ-VAEs in Section 5.1 and VQGANs in Section 5.2, codebook utilization and reconstruction metrics are significantly improved; however in this case, the quantization error is roughly the same.

## 6 Limitations

A limitation of the rotation trick can arise when the encoder outputs or codebook vectors are forced to be close to 0 norm (i.e., $\|e\| \approx 0$ or $\|q\| \approx 0$). In this case, the angle between $e$ and $q$ may be obtuse. When this happens, the rotation trick will "over-rotate" the gradient $\nabla_q \mathcal{L}$ as it is transported from $q$ to $e$ so that $\nabla_q \mathcal{L}$ and $\nabla_e \mathcal{L}$ now point in different directions (i.e. the cosine of the angle between $\nabla_e \mathcal{L}$ and $\nabla_q \mathcal{L}$ will be negative). An example is visualized in Figure 6.

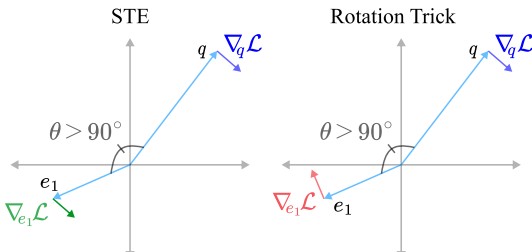

Figure 6: Illustration of the rotation trick "over-rotating" vectors when the angle between $e_1$ and $q$ is obtuse.

This is undesirable because—when the angle between $e$ and $q$ is obtuse—the rotation trick will violate the assumption that when $e \approx q$, $\nabla_q \mathcal{L} \approx \nabla_e \mathcal{L}$, and it will likely result in worse performance than VQ-VAEs trained with the STE. While obtuse angles between $e$ and $q$ are very unlikely—by design, the codebook vectors should be "angularly close" to the vectors that are mapped to them—however, if there is a restriction that forces codewords to have near 0 norm, then the rotation trick will likely perform worse than the STE.

## 7 Conclusion

In this work, we explore different ways to propagate gradients through the vector quantization layer of VQ-VAEs and find that preserving the angle—rather than the direction—between the codebook vector and gradient induces desirable effects for how points within the same codebook region are updated. These effects cause a substantial improvement in model performance. Across 11 different settings, we find that training VQ-VAEs with the rotation trick improves their reconstructions. For example, training one of the VQGANs used in latent diffusion with the rotation trick improves r-FID from 5.0 to 1.1 and r-IS from 141.5 to 200.2, reduces quantization error by two orders of magnitude, and increases codebook usage by 13.5x.

ACKNOWLEDGMENTS

We thank Henry Bosch, Benjamin Spector, Dan Biderman, Jordan Juravsky, Mayee Chen, Owen Dugan, Sabri Eyuboglu, and the Hazy Group as a whole for their invaluable feedback and help during revisions of this work. We gratefully acknowledge the support of NIH under No. U54EB020405 (Mobilize), NSF under Nos. CCF2247015 (Hardware-Aware), CCF1763315 (Beyond Sparsity), CCF1563078 (Volume to Velocity), and 1937301 (RTML); US DEVCOM ARL under Nos. W911NF-23-2-0184 (Long-context) and W911NF-21-2-0251 (Interactive Human-AI Teaming); ONR under Nos. N000142312633 (Deep Signal Processing); Stanford HAI under No. 247183; NXP, Xilinx, LETI-CEA, Intel, IBM, Microsoft, NEC, Toshiba, TSMC, ARM, Hitachi, BASF, Accenture, Ericsson, Qualcomm, Analog Devices, Google Cloud, Salesforce, Total, the HAI-GCP Cloud Credits for Research program, the Stanford Data Science Initiative (SDSI), and members of the Stanford DAWN project: Meta, Google, and VMWare. The U.S. Government is authorized to reproduce and distribute reprints for Governmental purposes notwithstanding any copyright notation thereon. Any opinions, findings, and conclusions or recommendations expressed in this material are those of the authors and do not necessarily reflect the views, policies, or endorsements, either expressed or implied, of NIH, ONR, or the U.S. Government.

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

# A    APPENDIX

Table 5: Results for TimeSformer-VQGAN trained on BAIR and UCF-101 with 1024 codebook vectors. †: model suffers from codebook collapse and diverges. r-FVD is computed on the validation set.

| Approach | Dataset | Codebook Usage | Train Loss (↓) | Quantization Error (↓) | Valid Loss (↓) | r-FVD (↓) |
|---|---|---|---|---|---|---|
| TimeSformer[†] | BAIR | 0.4% | 0.221 | 0.03 | 0.28 | 1661.1 |
| TimeSformer w/ Rotation Trick | BAIR | 43% | 0.074 | 3.0e-3 | **0.074** | **21.4** |
| TimeSformer[†] | UCF-101 | 0.1% | 0.190 | 0.006 | 0.169 | 2878.1 |
| TimeSformer w/ Rotation Trick | UCF-101 | 30% | 0.111 | 0.020 | **0.109** | **229.1** |

## A.1    VIDEO EVALUATION

Expanding our analysis beyond the image modality, we evaluate the effect of the rotation trick on video reconstructions from the BAIR Robot dataset (Ebert et al., 2017) and from the UCF101 action recognition dataset (Soomro, 2012). We follow the quantization paradigm used by ViT-VQGAN, but replace the ViT with a TimeSformer (Bertasius et al., 2021) video model. Due to compute limitations, both encoder and decoder follow a relatively small TimeSformer model: 8 layers, 256 hidden dimensions, 4 attention heads, and 768 MLP hidden dimensions. A complete description of the architecture, training settings, and hyperparameters are provided in Appendix A.10.4.

**Results.** Table 5 shows our results. For both datasets, training a TimeSformer-VQGAN model with the STE results in codebook collapse. We explored several different hyperparameter settings; however in all cases, codebook utilization drops to almost 0% within the first several epochs. On the other hand, models trained with the rotation trick do not exhibit any training instability and produce high quality reconstructions as indicated by r-FVD (Unterthiner et al., 2018). Several non-cherry picked video reconstructions are displayed in Appendix A.10.4.

## A.2    NON-CONVEX SYNTHETIC EXAMPLE

To supplement our analysis in Section 4.3, we include a numerical simulation of vector quantization for minimizing Himmelblau's function (Figure 7) across 100 gradient updates for the STE and rotation trick gradient estimators to highlight the differences in their behaviors. Our simulation uses an EMA with a decay rate of 0.8 as described in Van Den Oord et al. (2017) to update the codebook vectors and a learning rate of $1e-3$ to update the pre-quantized points. Points for both the STE and the rotation trick simulation use the same random initialization for both codewords and pre-quantized vectors. The only difference is whether the STE or the rotation trick is used as the gradient estimator through the vector quantization operation.

Figure 8 visualizes our results after 33, 66, and 100 gradient updates. The orange circles represent codebook vectors, the green dots the initial points, and the blue dots the updated points. Contour lines are drawn in each diagram to indicate regions of equal loss, with blue representing regions of low loss and red indicating regions of high loss. Similar to our findings in Section 5, we see that the rotation trick clusters points more tightly around each codebook vector when compared to the STE, resulting in lower distortion. Moreover, the codebook vectors more rapidly converge to the four equal local minima in Himmelblau's function, resulting in a lower objective function value when averaged across all points.

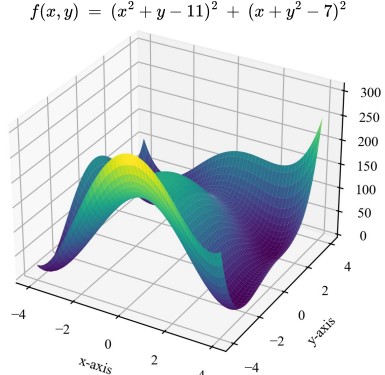

Figure 7: Loss surface for Himmelblau's function. Himmelblau's function has four equal local minima: $f(3.0, 2.0) = 0.0$, $f(-2.8.., 3.1...) = 0.0$, $f(-3.7.., -3.2..) = 0.0$, and $f(3.5.., -1.8..) = 0.0$.

## A.3    HESSIAN APPROXIMATION AND EXACT GRADIENT ANALYSIS

In this section, we expand our analysis in Section 3 and offer some intuition for why using exact gradients, or a Hessian approximation of the exact gradients, may convey undesirable characteristics. We begin by showing the Hessian approximates the exact gradient up to second order term with a

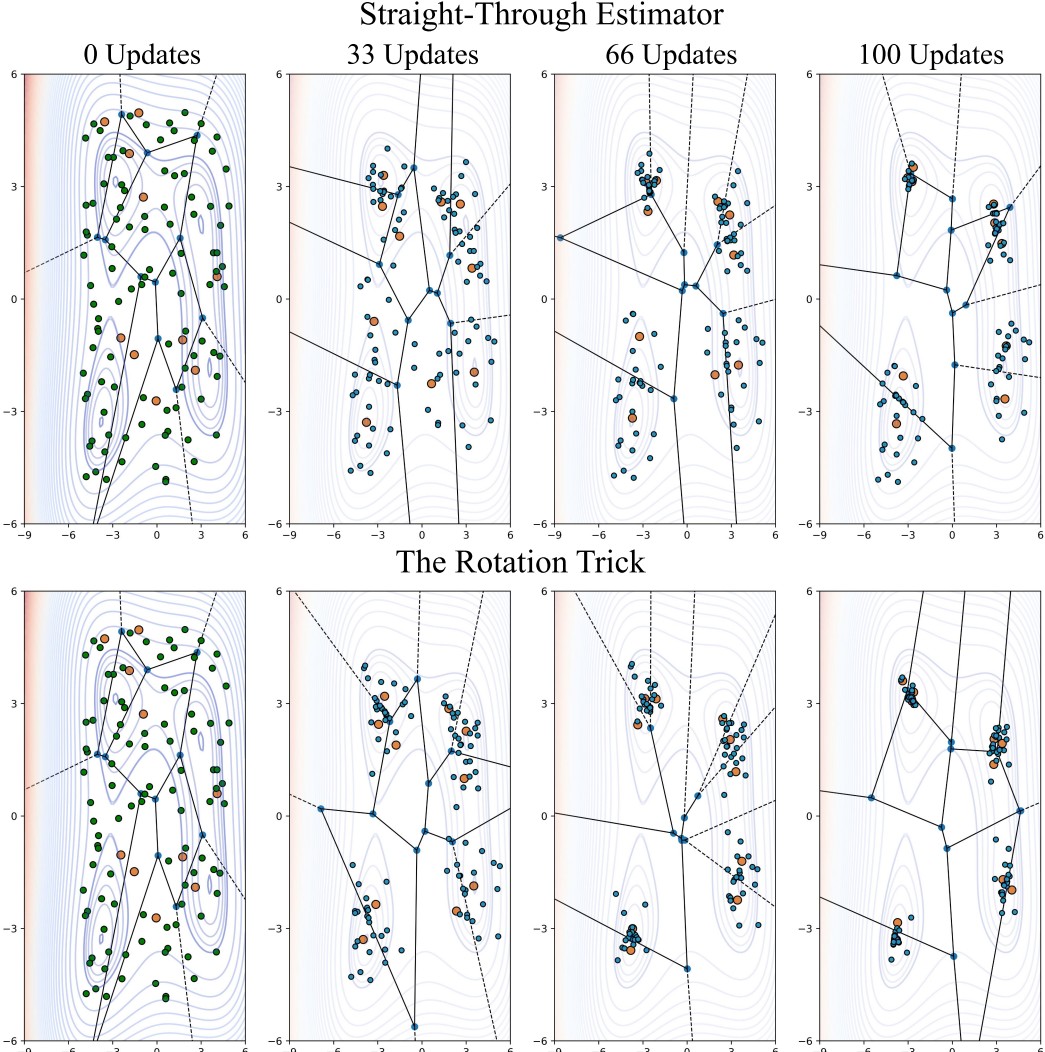

Figure 8: Synthetic experiment for minimizing Himmelblau's function with vector quantization using the STE gradient estimator (top row) and the rotation trick (bottom row). The rotation trick more quickly converges to these minima and achieves substantively lower distortion between codewords and pre-quantized points.

Taylor series expansion. We can write the loss $\mathcal{L}_e$ exactly as an infinite series of around $q$:

$$\mathcal{L}_e = \mathcal{L}_q + (\nabla_q\mathcal{L})^T(e-q) + \frac{1}{2}(e-q)^T(\nabla_q^2\mathcal{L})(e-q) + \frac{1}{6}(e-q)^T\nabla_q^3\mathcal{L}(e-q,e-q) + \dots.$$

so that the loss computed by the Hessian approximation differs from the loss computed with the exact gradients method by the remainder term from truncating the Taylor series expansion after the second term:

$$\{\mathcal{L}_e\}_{\text{Hessian}} = \mathcal{L}_q + (\nabla_q\mathcal{L})^T(e-q) + \frac{1}{2}(e-q)^T(\nabla_q^2\mathcal{L})(e-q)$$

When differentiating both of these losses to compute the gradients, the difference between the exact gradient update and the Hessian update is:

$$\frac{\partial\mathcal{L}_e}{\partial e} - \{\frac{\partial\mathcal{L}_e}{\partial e}\}_{\text{Hessian}} = \frac{\partial}{\partial e}\mathcal{O}(\|e-q\|^3)$$

where

$$\mathcal{O}(\|e-q\|^3) = \frac{1}{6}(e-q)^T\nabla_q^3\mathcal{L}(e-q,e-q) + \dots$$

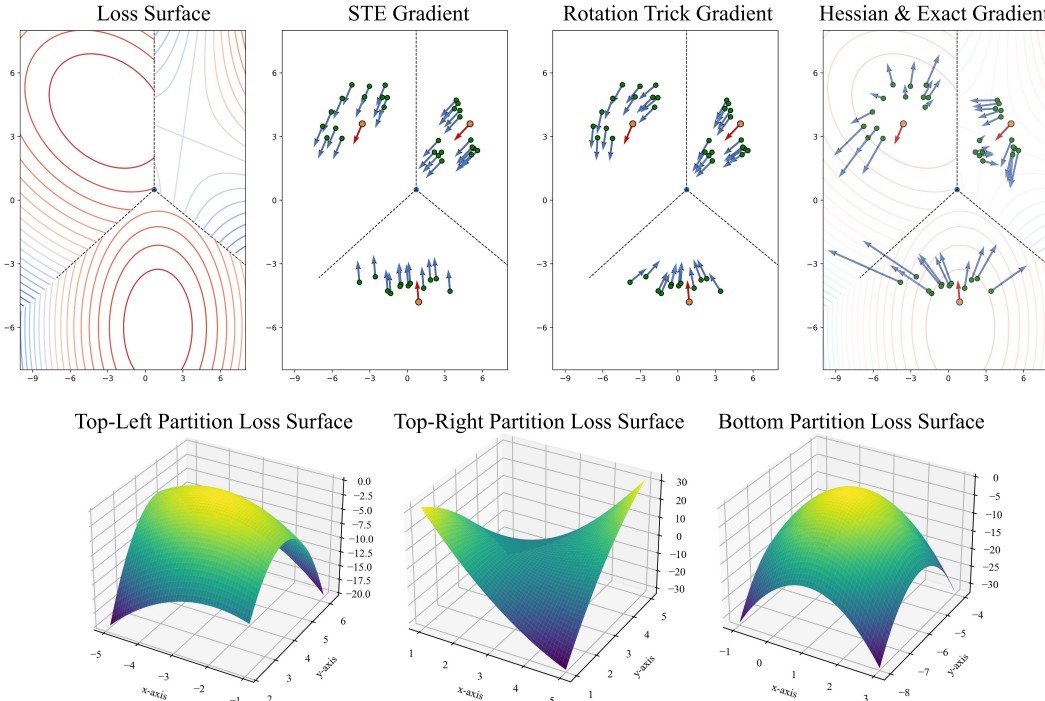

Figure 9: Examples of how the gradient can change due to the presence of negative curvature or an indefinite Hessian. As the loss in each partition is quadratic, the exact gradient will equal the Hessian approximation. Notice that when $q \approx e$, $\nabla_q \mathcal{L} \approx \nabla_e \mathcal{L}$ for both the STE and the rotation trick. As the Hessian approximation and exact gradients use the curvature of the loss surface to move $\nabla_q \mathcal{L}$ from $q$ to $e$, the direction of the gradient can change substantively, even when $q \approx e$.

The Hessian idea described in Section 3 approximates the exact gradients to the encoder as if quantization did not occur, i.e. it approximates the gradient used to update the encoder in the original AutoEncoder (Hinton & Zemel, 1993) model.

We now explore some instances where the exact gradients, or their Hessian approximation, may produce undesirable behavior in vector quantization. An inductive bias (Baxter, 2000) for vector quantization to work well is that when $e$ is "close" to $q$, their gradients are also "close", i.e. if $e \approx q$ then $\nabla_e \mathcal{L} \approx \nabla_q \mathcal{L}$. Intuitively, if the distortion between $e$ and $q$ is small—i.e. $q$ is a very good codeword for $e$—then these points should move together during a gradient update. If they do not, the distortion would increase.

This assumption holds for both the STE and Rotation Trick gradients; however, it can be violated by the Hessian approximation or the exact gradient approaches, especially when the curvature around $q$ is negative or the Hessian is indefinite and forms a saddle point.

Figure 9 illustrates three such cases. As both the STE and Rotation Trick do not use the loss surface to move $\nabla_q \mathcal{L}$ from $q$ to $e$, when $q \approx e$, $\nabla_q \mathcal{L} \approx \nabla_e \mathcal{L}$. However, approaches that use the curvature around $q$, such as the Hessian approximation or exact gradients, to either find or approximate the loss at $e$ can have $\nabla_e \mathcal{L}$ point in a very different direction from $\nabla_q \mathcal{L}$, even when $q$ is close to $e$. The top-left and bottom partitions of Figure 9 scatter the gradients as they move from $q$ to the points in these partitions due to negative curvature. A similar effect occurs in the top-right partition of Figure 9 due to the presence of a saddle point.

## A.4 BEHAVIOR AWAY FROM THE ORIGIN

Unlike the STE, the rotation trick is not invariant to the location of the origin. In this section, we explore this characteristic and its effect on how points within the same Voronoi region are updated. For example, suppose each codebook vector and encoder output in Figure 4 were shifted by some

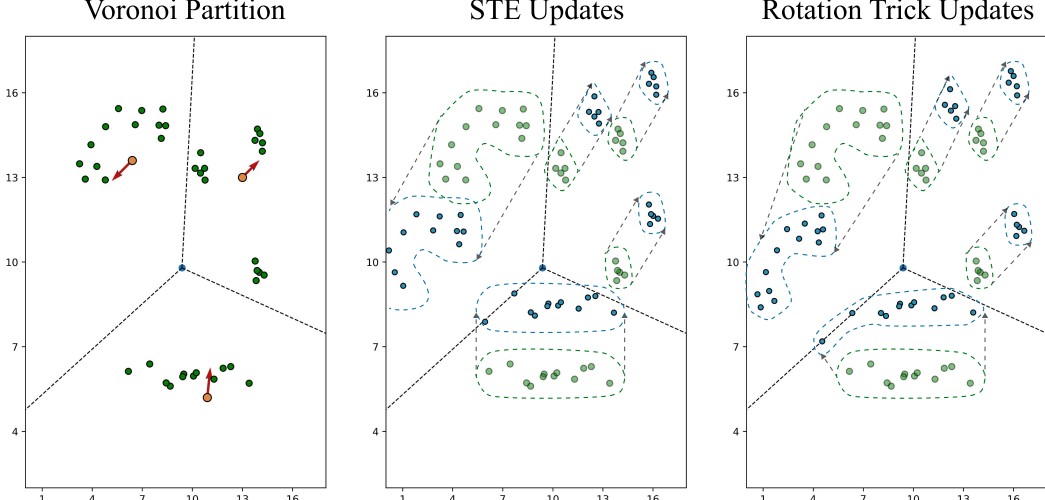

Figure 10: Depiction of how points within the same codebook region change after a gradient update (red arrow) at the codebook vector (orange circle) when all points are far from the origin. The STE is invariant to the this translation; however as the angle between $e$ and $q$ decreases as these vectors translated away from the origin, the effect of the rotation trick will decrease. In the limit, the rotation trick reduces to the STE.

constant vector so that each now has all positive components. How would this affect the rotation trick's gradient estimator?

Consider one codebook vector $q$ and one encoder output $e$ separated by angle $\theta$. We define $\hat{q} = q + d$ and $\hat{e} = e + d$ where $d$ is some large displacement vector. Let $\hat{\theta}$ be the angle between $\hat{q}$ and $\hat{e}$. We visualize this example in Figure 11. From the law of cosines:

$$\|q - e\|^2 = \|q\|^2 + \|e\|^2 - 2\|q\|\|e\|\cos(\theta)$$

and

$$\|\hat{q} - \hat{e}\|^2 = \|q - e\|^2 = \|\hat{q}\|^2 + \|\hat{e}\|^2 - 2\|\hat{q}\|\|\hat{e}\|\cos\left(\hat{\theta}\right)$$

Substituting, we find that

$$\cos\left(\hat{\theta}\right) = \frac{\|q\|^2 + \|e\|^2 - 2\|q\|\|e\|\cos(\theta) - \|q + d\|^2 - \|e + d\|^2}{-2\|q + d\|\|e + d\|}$$

and consider the case when $\hat{q}$ and $\hat{e}$ are far from the origin, i.e., $\|d\| >> \|q\|, \|e\|$. Then we have:

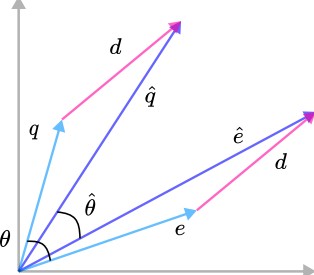

Figure 11: Illustration of codebook and encoder output shifted away from the origin by a constant vector $d$. The angle after the shift is smaller than the angle before the shift: $\hat{\theta} < \theta$.

$$\cos\left(\hat{\theta}\right) \approx \frac{-2\|d\|^2}{-2\|d\|^2} = 1$$

So as $d \to \infty$, $\hat{\theta} \to 0$. This implies that $\frac{\|\hat{q}\|}{\|\hat{e}\|} \to 1$ and $\hat{R} \to I$, which is exactly the STE update. As points move away from the origin, the rotation trick smoothly transforms into the STE.

We visualize an example of this effect in Figure 10, where each point from Figure 4 is translated by positive ten along each dimension. As illustrated above, the effect for the "push" gradient in the top-right quadrant remains but it's effect is reduced, i.e., more similar to the STE update. The top-left partition becomes a "pull" because the gradient now points towards the origin, so points within this region move closer together. Finally, the gradient in the bottom region no longer points towards the origin, but is now more orthogonal to the codebook vector. As a result, we see more of a rotation applied to the points in this region than the contraction that is depicted in Figure 4.

## A.5 HOUSEHOLDER REFLECTION TRANSFORMATION

For any given $e$ and $q$, the rotation $R$ that aligns $e$ with $q$ in the plane spanned by both vectors can be efficiently computed with Householder matrix reflections.

**Definition 1** (Householder Reflection Matrix). *For a unit norm vector $a \in \mathbb{R}^d$, $I - 2aa^T \in \mathbb{R}^{d \times d}$ is reflection matrix across the subspace (hyperplane) orthogonal to $a$.*

**Remark 1.** *Let $a, b \in \mathbb{R}^d$ that define hyperplanes $a^\perp$ and $b^\perp$ respectively. Then a reflection across $a^\perp$ followed by a reflection across $b^\perp$ is a rotation of $2\theta$ in the plane spanned by $a, b$ where $\theta$ is the angle between $a, b$.*

**Remark 2.** *Let $a, b \in \mathbb{R}^d$ with $\|a\| = \|b\| = 1$. Define $c = \frac{a+b}{\|a+b\|}$ as the vector half-way between $a$ and $b$ so that $\angle(a, b) = \theta$ and $\angle(a, c) = \angle(b, c) = \frac{\theta}{2}$. From Definition 1, $(I - 2cc^T)$ encodes a reflection across $c^\perp$ and $(I - 2bb^T)$ encodes a reflection across $b^\perp$. From Remark 1, $(I - 2bb^T)(I - 2cc^T)$ then corresponds to a rotation of $2(\frac{\theta}{2}) = \theta$ in the plane spanned by $b$ and $c$. As the $\text{span}(b, c) = \text{span}(a, b)$, $(I - 2bb^T)(I - 2cc^T)$ corresponds to a rotation of $\theta$ in the plane spanned by $a$ and $b$. Therefore, $(I - 2bb^T)(I - 2cc^T)a = b$.*

Returning to vector quantization with $q = [\frac{\|q\|}{\|e\|} R]e$, we can write $R$ as the product of two Householder reflection matrices that rotates $e$ to $q$ in the plane spanned between them. Without loss of generality, assume $e$ and $q$ are unit norm, and let $\theta$ be the angle between $e$ and $q$. Setting $r = \frac{e+q}{\|e+q\|}$ and simplifying yields:

$$
\begin{aligned}
R &= (I - 2qq^T)(I - 2rr^T) \\
&= I - 2qq^T - 2rr^T + 4qq^T rr^T \\
&= I - 2qq^T - 2rr^T + 4q\left[q^T r\right] r^T \\
&= I - 2qq^T - 2rr^T + 4q\left[q^T \frac{e+q}{\|e+q\|}\right] r^T \\
&= I - 2qq^T - 2rr^T + 4q\left[\frac{q^T e + q^T q}{\|e+q\|}\right] r^T \\
&= I - 2qq^T - 2rr^T + 4q\left[\frac{\|q\|\|e\| \cos\theta + \|q\|\|q\|}{\|e+q\|}\right] r^T \\
&= I - 2qq^T - 2rr^T + 4q\left[\frac{\cos\theta + 1}{\|e+q\|}\right] r^T \\
&= I - 2qq^T - 2rr^T + 4q\left[\frac{\|e+q\|^2}{2\|e+q\|}\right] r^T \\
&= I - 2qq^T - 2rr^T + \frac{4\|e+q\|^2}{2\|e+q\|} qr^T \\
&= I - 2qq^T - 2rr^T + \frac{4\|e+q\|^2}{2\|e+q\|^2} q(e+q)^T \\
&= I - 2qq^T - 2rr^T + 2qe^T + 2qq^T \\
&= I - 2rr^T + 2qe^T
\end{aligned}
$$

## A.6 PROOF THE ROTATION TRICK PRESERVES ANGLES

For encoder output $e$ and corresponding codebook vector $q$, we provide a formal proof that the rotation trick preserves the angle between $\nabla_q \mathcal{L}$ and $q$ as $\nabla_q \mathcal{L}$ moves to $e$. Unlike the notation in the main text, which assumes $q \in \mathbb{R}^{d \times 1}$, we use batch notation in the following proof to illustrate how the rotation trick works when training neural networks. Specifically, $q \in \mathbb{R}^{b \times d}$ and $R \in \mathbb{R}^{b \times d \times d}$ where $b$ is the number of examples in a batch and $d$ is the dimension of the codebook vector.

**Remark 3.** *The angle between $q$ and $\nabla_q \mathcal{L}$ is preserved as $\nabla_q \mathcal{L}$ moves to $e$.*

*Proof.* With loss of generality, suppose $\|e\| = \|q\| = 1$. Then we have

$$q = eR^T$$

$$\frac{\partial q}{\partial e} = R$$

The gradient at $e$ will then equal:

$$\nabla_e \mathcal{L} = \nabla_q \mathcal{L} \left[\frac{\partial q}{\partial e}\right]$$

$$= \nabla_q \mathcal{L} [R]$$

Let $\theta$ be the angle between $q$ and $\nabla_q \mathcal{L}$ and $\phi$ be the angle between $e$ and $\nabla_q \mathcal{L}$. Via the Euclidean inner product, we have:

$$\|\nabla_q \mathcal{L}\| \cos \theta = q [\nabla_q \mathcal{L}]^T$$

$$= eR^T [\nabla_q \mathcal{L}]^T$$

$$= e [\nabla_q \mathcal{L} R]^T$$

$$= e [\nabla_e \mathcal{L}]^T$$

$$= \|\nabla_q \mathcal{L}\| \cos \phi$$

so $\theta = \phi$ and the angle between $q$ and $\nabla_q \mathcal{L}$ is preserved as $\nabla_q \mathcal{L}$ moves to $e$. $\qquad\square$

### A.7 TREATING $R$ AND $\frac{\|q\|}{\|e\|}$ AS CONSTANTS

In the rotation trick, we treat $R$ and $\frac{\|q\|}{\|e\|}$ as constants and detached from the computational graph during the forward pass of the rotation trick. In this section, we explain why this is the case.

The rotation trick computes the input to the decoder $\tilde{q}$ after performing a non-differentiable codebook lookup on $e$ to find $q$. It is defined as:

$$\tilde{q} = \frac{\|q\|}{\|e\|} Re$$

As shown in Section 4, $R$ is a function of both $e$ and $q$. However, using the quantization function $\mathcal{Q}(e) = q$, we can rewrite both $\frac{\|q\|}{\|e\|}$ and $R$ as a single function of $e$:

$$f(e) = \frac{\|\mathcal{Q}(e)\|}{\|e\|} \left[I - 2 \left[\frac{e + \mathcal{Q}(e)}{\|e + \mathcal{Q}(e)\|}\right] \left[\frac{e + \mathcal{Q}(e)}{\|e + \mathcal{Q}(e)\|}\right]^T + 2\mathcal{Q}(e)e^T\right]$$

$$= \frac{\|q\|}{\|e\|} R$$

The rotation trick then becomes

$$\tilde{q} = f(e)e$$

and differentiating $\tilde{q}$ with respect to $e$ gives us:

$$\frac{\partial \tilde{q}}{\partial e} = f'(e)e + f(e)$$

However, $f'(e)$ cannot be computed as it would require differentiating through $\mathcal{Q}(e)$, which is a non-differentiable codebook lookup. We therefore drop this term and use only $f(e)$ as our approximation of the gradient through the vector quantization layer: $\frac{\partial \tilde{q}}{\partial e} = f(e)$. This approximation conveys more information about the vector quantization operation than the STE, which sets $\frac{\partial \tilde{q}}{\partial e} = I$.

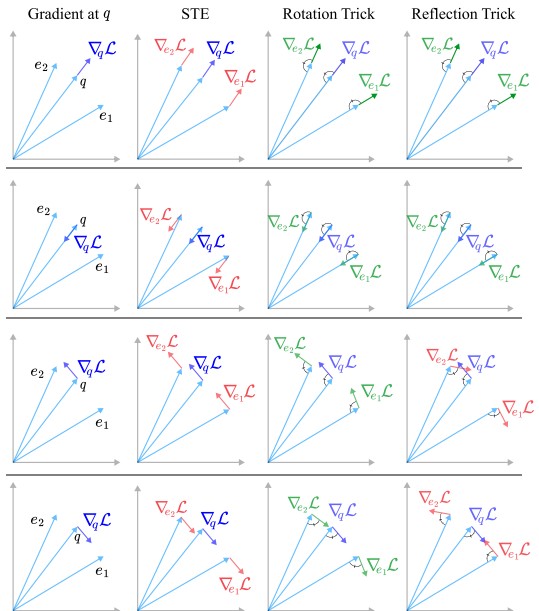

Figure 12: Illustration of how the gradient at $q$ moves to $e$ via the STE, the rotation trick, and the reflection trick. The reflection trick matches the behavior of the rotation trick when the gradient $\nabla_q \mathcal{L}$ is parallel to $q$. However, it will reverse the components of the gradients orthogonal to $q$ for points in q's partition. This effect is illustrated in the bottom two rows of the rightmost column.

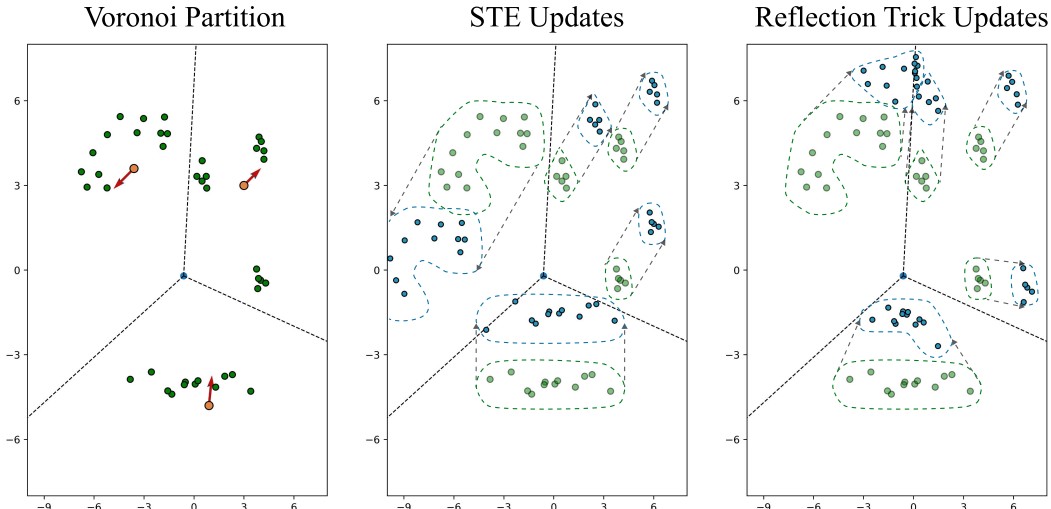

Figure 13: Depiction of how points within the same codebook region change after a gradient update (red arrow) at the codebook vector (orange circle). The STE applies the same update to each point in the same region. The reflection trick (Appendix A.8) modifies the update based on the location of each point with respect to the codebook vector. Note the top-left region of the reflection trick update, where the points actually move in the opposite direction of the gradient update.

## A.8 THE REFLECTION TRICK

One may also use a single reflection to align $e$ to $q$, rather than a rotation. For instance, using the notation from Appendix A.5, setting $r = \frac{e-q}{\|e-q\|}$ and reflecting across the plane orthogonal to this vector via the Householder reflection $(I - 2rr^T)$ will reflect $e$ to $q$. We denote this reflection as $\tilde{R}$ so that $\tilde{q} = \frac{\|q\|}{\|e\|} \tilde{R} e$. We call this approach "the reflection trick."

The reflection trick can result in undesirable behavior during the backward pass. While it replicates the rotation trick when $\nabla_q \mathcal{L}$ is parallel to $q$, as illustrated in the top two rows of Figure 12 and the top-right and bottom regions of Figure 13, it reflects orthogonal components of the gradient across the hyperplane orthogonal to $e - q$ so that these components are reversed. Simply, if the quantized gradient points "left" then the reflected gradient will point "right", and vice-versa. This behavior is undesirable for points with low distortion, $e \approx q$, because it will cause $e$ to move away from $q$ along the components of the gradient orthogonal to $q$, thereby increasing distortion for two points that are a "good match". The top-left partition of Figure 13 illustrates one such example. In this case, the gradient pushes the codebook vector "left" while the points in this region are pushed in the opposite direction of the gradient.

We evaluate this effect experimentally following the VQ-VAE evaluation paradigm from Table 1 and the VQGAN evaluation paradigm from Table 3. While we did not train these models to completion due to GPU resource limitations, both paradigms exhibited poor convergence when trained with the reflection trick. Specifically, after one epoch, the validation loss was approximately 3x higher than the rotation trick for both 8192 and 16384 codebook VQGANs in Table 3. For the Euclidean codebook model with latent Shape $64 \times 64 \times 3$ in Table 1, the validation loss was approximately 2x higher than the rotation trick after 15 epochs.

## A.9 GRADIENT NORM SCALING IN THE ROTATION TRICK

In this section, we analyze the effect of the $\frac{\|q\|}{\|e\|}$ term in the rotation trick. While this norm rescaling is necessary to transform $e$ into $q$ during the forward pass, one could avoid the multiplicative factor by instead formulating the rotation trick as:

$$\tilde{q} = \underbrace{R}_{\text{constant}} e + \underbrace{(q - Re)}_{\text{constant}}$$

A possible benefit of this latter formulation is that $\frac{\partial q}{\partial e} = R$, an orthogonal transformation with determinant one that does not shrink or expand space by a factor of $\frac{\|q\|}{\|e\|}$. In this section, we analyze the differences between these two approaches and formulate both as specific instantiations of a more general family of rotation-based gradient approximations.

### A.9.1 COMPARISON BETWEEN $\frac{\|q\|}{\|e\|}$ AND $(q - Re)$

An inductive bias of vector quantization is that when $e \approx q$, then $\nabla_e \mathcal{L} \approx \nabla_q \mathcal{L}$. Simply, when the distortion between $e$ and $q$ is small, the gradient for both $e$ and $q$ should be approximately the same. However when $\|e\| \approx 0$ and a Euclidean metric is used to determine the closest codebook vector, the angle between $e$ and $q$ can be obtuse as illustrated in Figure 6. In this instance, the rotation trick will cause the gradient $\nabla_e \mathcal{L}$ to "over-rotate" and point away from $\nabla_q \mathcal{L}$.

Using a grad scaling of $\frac{\|q\|}{\|e\|}$ can fix this. When $\|e\| \approx 0$ and $\|e\| < \|q\|$, the norm of the gradient will be scaled up to push $e$ away from the origin. Pushing $e$ away from the origin makes the angle between $e$ and $q$ more of a factor when computing the Euclidean distance:

$$\|e - q\| = \sqrt{\|e\|^2 + \|q\|^2 - 2\|e\|\|q\|\cos\theta}$$

so $e$ is more likely to map to a different $q$ that forms an acute angle with it as $\|e\|$ increases.

Now consider if $\|q\| \approx 0$ and $\|e\| > \|q\|$. When this occurs, the update to $e$ will vanish because $\frac{\|q\|}{\|e\|} \approx 0$. This behavior may also be desirable because when $q$ is close to the origin, there's a higher likelihood the angle between $e$ and $q$ would be obtuse.

We also explore this factor in ablation experiments for VQ-VAEs and VQGANs. Table 6 mirrors Table 1 and summarizes our findings for VQ-VAEs while Table 7 mirrors Table 3 and summarizes our findings for the VQGANs used in latent diffusion. In Table 6, we do not observe a difference between using $\tilde{q} = \frac{\|q\|}{\|e\|} Re$ and $\tilde{q} = Re + (q - Re)$. However, for the VQGAN results in Table 7, we find that using the grad scaling factor modestly improves performance.

Table 6: Comparison of the rotation trick using $\tilde{q} = \frac{\|q\|}{\|e\|} Re$ with using $\tilde{q} = Re + (q - Re)$ for VQ-VAE models. The experimental setting follows Table 1.

| Rotation Trick Function | Training Metrics | | | Validation Metrics | | |
|---|---|---|---|---|---|---|
| | Codebook Usage ($\uparrow$) | Rec. Loss ($\downarrow$) | Quantization Error ($\downarrow$) | Rec. Loss ($\downarrow$) | r-FID ($\downarrow$) | r-IS ($\uparrow$) |
| Codebook Lookup: Euclidean & Latent Shape: $64 \times 64 \times 3$ & Codebook Size: 8192 | | | | | | |
| $\frac{\|q\|}{\|e\|} Re$ | 99% | 0.028 | 1.4e-4 | 0.030 | 16.5 | 106.3 |
| $Re - (q - Re)$ | 100% | 0.028 | 4.0e-4 | 0.030 | 16.5 | 106.1 |

Table 7: Comparison of the rotation trick using $\tilde{q} = \frac{\|q\|}{\|e\|} Re$ with using $\tilde{q} = Re + (q - Re)$ for VQGAN models. The models with codebook size were stopped after 2 epochs while the models with codebook size 16384 were stopped after 3 epochs.

| Rotation Trick Function | Latent Shape | Codebook Size | Codebook Usage | Quantization Error ($\downarrow$) | Valid Loss ($\downarrow$) | r-FID ($\downarrow$) | r-IS ($\uparrow$) |
|---|---|---|---|---|---|---|---|
| $\frac{\|q\|}{\|e\|} Re$ | $64 \times 64 \times 3$ | 8192 | 45% | 4.0e-4 | 0.161 | 0.46 | 225.0 |
| $Re - (q - Re)$ | $64 \times 64 \times 3$ | 8192 | 28% | 1.5e-3 | 0.183 | 0.6 | 220.0 |
| $\frac{\|q\|}{\|e\|} Re$ | $32 \times 32 \times 4$ | 16384 | 18% | 3.3e-4 | 0.292 | 1.5 | 196.1 |
| $Re - (q - Re)$ | $32 \times 32 \times 4$ | 16384 | 13% | 9.4e-4 | 0.292 | 1.5 | 191.5 |

### A.9.2 GENERAL FAMILY OF ROTATION-BASED GRADIENT ESTIMATORS

Generalizing the additive and multiplicative formulations of the rotation trick, we formulate both as specific instantiations of a more general family:

$$\tilde{q} = \gamma(e)Re + (q - \gamma(e)Re)$$

where $\gamma(e)$ determines the multiplicative scaling factor. For $\tilde{q} = \frac{\|q\|}{\|e\|} Re$, $\gamma(e) = \frac{\|q\|}{\|e\|}$ and for $\tilde{q} = Re + (q - Re)$, $\gamma(e) = 1$. However, one can explore other scaling factors such as

$$\gamma(e) = \frac{1}{8\|q - e\|^2}$$

We visualize the gradient fields for different formulations of $\gamma(e)$ in Figure 14.

It is almost certain that other formulations of $\gamma(e)$ from the ones we explore in this work would improve the training dynamics or performance of VQ-VAEs. In particular, *a priori* fixing $\gamma(e)$ to satisfy an inductive bias or developing an adaptive scaling factor that dynamically sets $\gamma(e)$ similar to the functions that adapt task weights in multi-task learning throughout training (Kendall et al., 2018; Chen et al., 2018) are exciting directions for future work.

### A.10 TRAINING SETTINGS

We detail the training settings used in our experimental analysis in Section 5. While a text description can be helpful for understanding the experimental settings, our released code should be referenced to fully reproduce the results presented in this work.

### A.10.1 VQ-VAE EVALUATION.

Table 8 summarizes the hyperparameters used for the experiments in Section 5.1. For the encoder and decoder architectures, we use the Convolutional Neural Network described by Esser et al. (2021). The hyperparameters for the cosine similarity codebook lookup follow from Yu et al. (2021) and the hyperparameters for the Euclidean distance codebook lookup follow from the default values set in the Vector Quantization library from https://github.com/lucidrains/vector-quantize-pytorch. All models replace the codebook loss with the exponential moving average described in Van Den Oord et al. (2017) with decay = 0.8. The notation for both encoder and decoder architectures is adapted from Esser et al. (2021).

For the Gumbel VQ-VAE baseline, we follow the implementation of https://github.com/karpathy/deep-vector-quantization and use the suggested schedule to attenuate the softmax temperature from 1.0 to $\frac{1}{16}$ over the course of training. Aside from the difference in quantization, i.e. deterministic

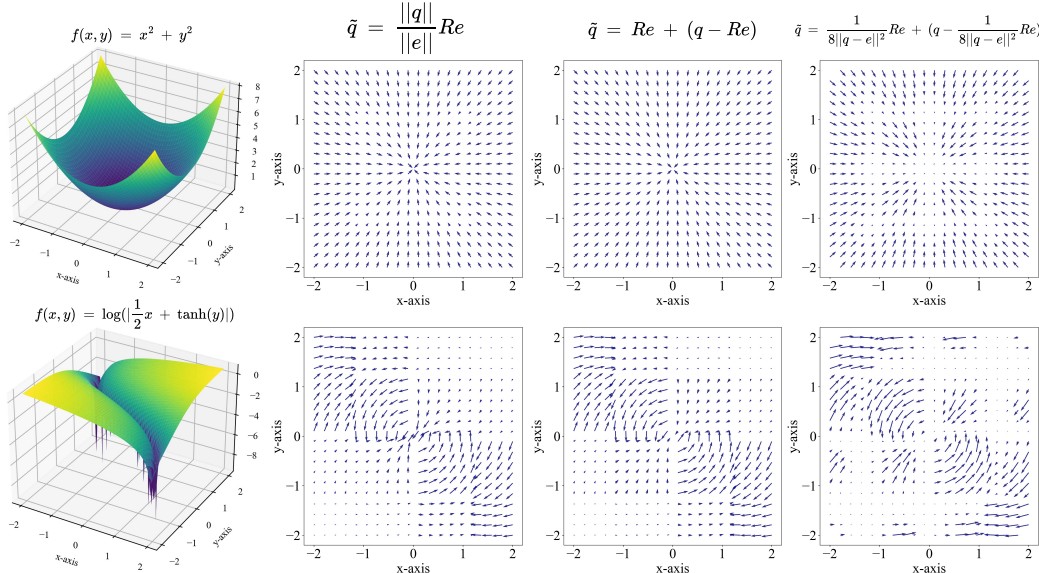

Figure 14: Visualization of how different choices of $\gamma(e)$ in the rotation trick affect the gradient field for (top) $f(x, y) = x^2 + y^2$ and (bottom) $f(x, y) = \log\left(|\frac{1}{2}x + \tanh(y)|\right)$. To prevent cluttered visualizations, the maximum and minimum gradient norms are capped within the gradient field.

Table 8: Hyperparameters for the experiments in Table 1. (1024, 32) indicates a model trained with a codebook size of 1024 and codebook dimension of 32. Similarly, (8192, 3) indicates a model trained with codebook size of 8192 and codebook dimension of 3.

|  | Cosine Similarity Lookup | | Euclidean Lookup | |
| --- | --- | --- | --- | --- |
|  | (1024, 32) | (8192, 3) | (1024, 32) | (8192, 3) |
| Input size | $256 \times 256 \times 3$ | $256 \times 256 \times 3$ | $256 \times 256 \times 3$ | $256 \times 256 \times 3$ |
| Latent size | $16 \times 16 \times 32$ | $64 \times 64 \times 3$ | $16 \times 16 \times 32$ | $64 \times 64 \times 3$ |
| $\beta$ (commitment loss coefficient) | 1.0 | 1.0 | 1.0 | 1.0 |
| encoder/decoder channels | 128 | 128 | 128 | 128 |
| encoder/decoder channel mult. | [1, 1, 2, 2, 4] | [1, 2, 4] | [1, 1, 2, 2, 4] | [1, 2, 4] |
| [Effective] Batch size | 256 | 256 | 256 | 256 |
| Learning rate | $1 \times 10^{-4}$ | $1 \times 10^{-4}$ | $5 \times 10^{-5}$ | $5 \times 10^{-5}$ |
| Weight Decay | $1 \times 10^{-4}$ | $1 \times 10^{-4}$ | 0 | 0 |
| Codebook size | 1024 | 8192 | 1024 | 8192 |
| Codebook dimension | 32 | 3 | 32 | 3 |
| Training epochs | 25 | 20 | 25 | 20 |

versus stochastic, the architecture and optimization of the Gumbel VQ-VAE model are identical to the VQ-VAE baseline.

### A.10.2 VQGAN EVALUATION

Table 9 summarizes the hyperparameters for the VQGAN experiments in Section 5.2. For the Gumbel VQGAN model, we follow the default hyperparameters and settings from Rombach et al. (2022). Non-cherry picked reconstructions for the models trained in Table 2 and Table 3 are depicted in Figure 15. As indicated by the increased r-FID score, the reconstructions out by the VQGAN trained with the rotation trick appear to better reproduce the original image, especially fine details.

### A.10.3 VIT-VQGAN EVALUATION

Our experiments in Section 5.3 use the ViT-VQGAN implemented in the open source repository https://github.com/thuanz123/enhancing-transformers. The default hyperparameters follow those

VQGAN from Taming Transformers | VQGAN from Latent Diffusion

ImageNet | FFHQ & CelebA-HQ | ImageNet [f=8] | [f=4]

Orig STE ROT | Orig STE ROT | Orig STE ROT STE ROT

Figure 15: Non-cherry picked reconstructions for VQGAN results in Table 2 and Table 3. *ROT* is an abbreviation for the rotation trick.

Table 9: Hyperparameters for the experiments in Table 2 and Table 3. We implement the rotation trick in the open source https://github.com/CompVis/taming-transformers for the experiments in Table 2 and implement the rotation trick in https://github.com/CompVis/latent-diffusion for Table 3. In both settings, we use the default hyperparameters. †: 18 epochs for ImageNet and 50 epochs for FFHQ & CelebA-HQ.

|  | Table 2 VQGAN | Table 3 VQGAN | Table 3 VQGAN |
|---|---|---|---|
| Input size | $256 \times 256 \times 3$ | $256 \times 256 \times 3$ | $256 \times 256 \times 3$ |
| Latent size | $16 \times 16 \times 256$ | $64 \times 64 \times 3$ | $32 \times 32 \times 4$ |
| Codebook weight | 1.0 | 1.0 | 1.0 |
| Discriminator weight | 0.8 | 0.75 | 0.6 |
| encoder/decoder channels | 128 | 128 | 128 |
| encoder/decoder channel mult. | $[1, 1, 2, 2, 4]$ | $[1, 2, 4]$ | $[1, 2, 2, 4]$ |
| [Effective] Batch size | 48 | 16 | 16 |
| [Effective] Learning rate | $4.5 \times 10^{-6}$ | $4.5 \times 10^{-6}$ | $4.5 \times 10^{-6}$ |
| Codebook size | 1024 | 8192 | 16384 |
| Codebook dimensions | 256 | 3 | 4 |
| Training Epochs | $18/50^{\dagger}$ | 4 | 4 |

specified by Yu et al. (2021), and our experiments use the default architecture settings specified by the ViT small model configuration file.

We depict several reconstructions in Figure 16 and see that the ViT-VQGAN trained with the rotation trick is able to better replicate small details that the ViT-VQGAN trained with the STE misses. This is expected as the rotation trick drops r-FID from 29.2 to 11.2 as shown in Table 4.

### A.10.4   TIMESFORMER VIDEO EVALUATION

We use the Hugging Face implementation of the TimeSformer from https://huggingface.co/docs/transformers/en/model_doc/timesformer and the ViT-VQGAN vector quantization layer from https://github.com/thuanz123/enhancing-transformers. We loosely follow the hyperparameters listed in Yu et al. (2021) and implement a small TimeSformer encoder and decoder due to GPU VRAM constraints. We reuse the dataloading functions of both BAIR Robot Pushing and UCF101 dataloaders from Yan et al. (2021) at https://github.com/wilson1yan/VideoGPT. A complete description of the settings we use for the experiments in Appendix A.1 are listed in Table 11.

We also visualize the reconstructions for the TimeSformer-VQGAN trained with the rotation trick and the STE. Figure 17 shows the reconstructions for BAIR Robot Pushing, and Figure 18 shows the

Table 10: Hyperparameters for the experiments in Table 4.

|  | ViT-VQGAN Settings |
| --- | --- |
| Input size | $256 \times 256 \times 3$ |
| Patch size | 8 |
| Encoder / Decoder Hidden Dim | 512 |
| Encoder / Decoder MLP Dim | 1024 |
| Encoder / Decoder Hidden Depth | 8 |
| Encoder / Decoder Hidden Num Heads | 8 |
| Codebook Dimension | 32 |
| Codebook Size | 8192 |
| Codebook Loss Coefficient | 1.0 |
| Log Laplace loss Coefficient | 0.0 |
| Log Gaussian Coefficient | 1.0 |
| Perceptual loss Coefficient | 0.1 |
| Adversarial loss Coefficient | 0.1 |
| [Effective] Batch size | 32 |
| Learning rate | $1 \times 10^{-4}$ |
| Weight Decay | $1 \times 10^{-4}$ |
| Training epochs | 10 |

ViT-VQGAN

Orig STE ROT

Figure 16: Non-cherry picked reconstructions for ViT-VQGAN results in Table 4. *ROT* is an abbreviation for the rotation trick.

Original Video      STE Reconstructions      Rotation Trick Reconstructions

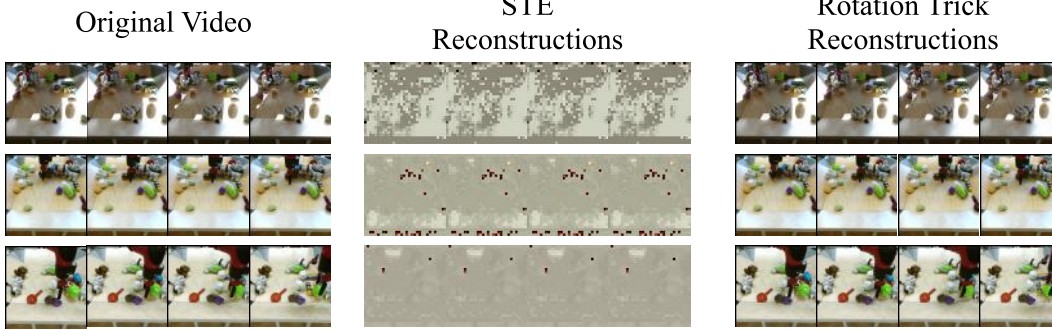

Figure 17: BAIR Robot Pushing reconstruction examples. While the model trains on 16 video frames at a time, we only visualize 4 at a time in this figure. The model trained with the STE undergoes codebook collapse, using 4 out of the 1024 codebook vectors for reconstruction and therefore crippling the information capacity of the vector quantization layer. On the other hand, the VQ-VAE trained with the rotation trick instead uses an average of 441 of the 1024 codebook vectors in each batch of 2 example videos.

reconstructions for UCF101. For both datasets, the model trained with the STE undergoes codebook collapse early into training. Specifically, it learns to only use $\frac{4}{1024}$ of the available codebook vectors for BAIR Robot Pushing and $\frac{2}{2048}$ for UCF101 in a batch of 2 input examples. Small manual tweaks to the architecture and training hyperparameters did not fix this issue.

In contrast, VQ-VAEs trained with the rotation trick do not manifest this training instability. Instead, codebook usage is relatively high—at 43% for BAIR Robot Pushing and 30% for UCF101—and the reconstructions accurately match the input, even though both encoder and decoder are very small video models.

### A.10.5 CREATION OF VORONOI REGION FIGURE

In this section, we describe the creation of Figure 4 as well as the other figures that use this format. For the top-right and bottom partitions, we fix the codebook to a set of preset values and sample pre-quantized points from four different Gaussian distributions. For the pre-quantized points in the top-left partition, we manually set them to form a crescent shape around the codeword.

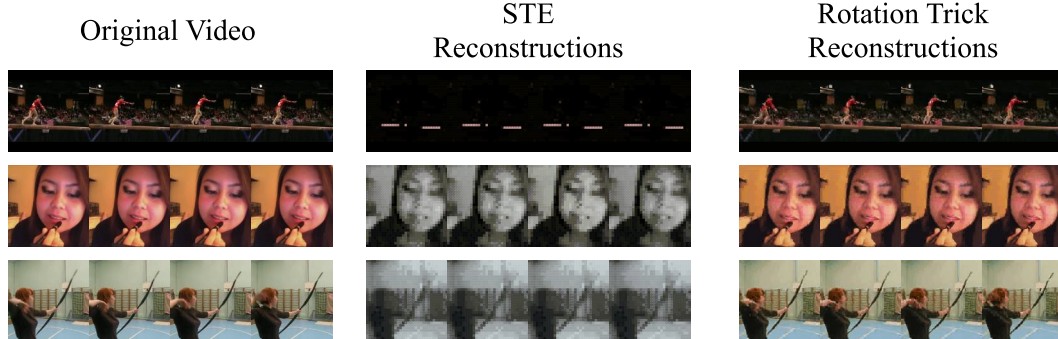

Figure 18: UCF-101 reconstruction examples. While the model trains on 16 video frames at a time, we only visualize 4 at a time in this figure. The model trained with the STE undergoes codebook collapse, using approximately 2 out of the 2048 codebook vectors for reconstruction and therefore crippling the information capacity of the vector quantization layer. The VQ-VAE trained with the rotation trick instead uses an average of 615 of the 2048 codebook vectors in each batch of 2 example videos.

Table 11: Hyperparameters for the experiments in Table 5. A TimeSformer (Bertasius et al., 2021) is used for the Encoder and Decoder architecture as implemented at https://huggingface.co/docs/transformers/en/model_doc/timesformer. The vector quantization layer between Encoder and Decoder follow from Yu et al. (2021) as implemented in https://github.com/thuanz123/enhancing-transformers.

|  | TimeSformer-VQGAN Settings | |
| --- | --- | --- |
|  | BAIR Robot Pushing | UCF101 Action Recognition |
| Input size | $16 \times 64 \times 64 \times 3$ | $16 \times 128 \times 128 \times 3$ |
| Patch size | 2 | 4 |
| Encoder / Decoder Hidden Dim | 256 | 256 |
| Encoder / Decoder MLP Dim | 768 | 768 |
| Encoder / Decoder Hidden Depth | 8 | 8 |
| Encoder / Decoder Hidden Num Heads | 4 | 4 |
| Codebook Dimension | 32 | 32 |
| Codebook Size | 1024 | 2048 |
| Codebook Loss Coefficient | 1.0 | 1.0 |
| Log Laplace loss Coefficient | 0.0 | 0.0 |
| Log Gaussian Coefficient | 1.0 | 1.0 |
| Perceptual loss Coefficient | 0.1 | 0.1 |
| Adversarial loss Coefficient | 0.1 | 0.1 |
| [Effective] Batch size | 24 | 20 |
| Learning rate | $1 \times 10^{-4}$ | $4.5 \times 10^{-6}$ |
| Weight Decay | $1 \times 10^{-4}$ | $1 \times 10^{-4}$ |
| Training epochs | 30 | 3 |

We similarly fix constant gradient vectors for each partition, and apply them to the pre-quantized points after transformation by the STE, i.e. simply moving the gradient to each pre-quantized point in the quantized region, or by the rotation trick, i.e. rotating the gradient based on the angle between the pre-quantized point and closest codebook vector and rescaling appropriately. We multiply the gradient by a small constant—the learning rate—and then apply the gradient to each pre-quantized point. We repeat the above 25 times, at each point re-computing the angle and magnitude between the pre-quantized point and the codebook vector for the rotation trick update. For simplicity, we do not update the codebook vectors themselves or recompute codebook regions throughout the numerical simulation.

## A.11 COMPARISON WITHIN GENERATIVE MODELING APPLICATIONS

Absent from our work is an analysis on the effect of VQ-VAEs trained with the rotation trick on down-stream generative modeling applications. We see this comparison as outside the scope of this

work and do not claim that improving reconstruction metrics, codebook usage, or quantization error in "Stage 1" VQ-VAE training will lead to improvements in "Stage 2" generative modeling applications.

While poor reconstruction performance will clearly lead to poor generative modeling, recent work (Yu et al., 2023) suggests that—at least for autoregressive modeling of codebook sequences with MaskGit (Chang et al., 2022)—the connection between VQ-VAE reconstruction performance and downstream generative modeling performance is non-linear. Specifically, increasing the size of the codebook past a certain amount will improve VQ-VAE reconstruction performance but make downstream likelihood-based geneative modeling of codebook vectors more difficult.

We believe this nuance may extend beyond MaskGit, and that the *desiderata* for likelihood-based generative models will likely be different than that for score-based generative models like diffusion. It is even possible that different preferences appear within the same class. For example, left-to-right autoregressive modeling of codebook elements with Transformers (Vaswani, 2017) may exhibit different preferences for Stage 1 VQ-VAE models than those of MaskGit.

These topics deserve a deep, and rich, analysis that we would find difficult to include within this work as our focus is on propagating gradients through vector quantization layers. As a result, we entrust the exploration of these questions to future work.

### A.12 GRADIENT ESTIMATORS AS PARALLEL TRANSPORT

In this section, we analyze the STE and the rotation trick through the lens of differential geometry, specifically as the parallel transport of the gradient $\nabla_q \mathcal{L}$ vector from the codeword $q$ to the encoder output $e$. For this analysis in this section, we only consider the rotational component $R_\theta$ of the rotation trick, not the rescaling by $\frac{\|q\|}{\|e\|}$.

#### A.12.1 BACKGROUND ON HYPERSPHERICAL COORDINATES

Hyperspherical coordinate systems are ubiquitous in applications of math and physics, where certain formulas become greatly simplified by parameterizing the location of points by the radius and angles to coordinate axes. An familiar instantiation of the hyperspherical coordinate system may be polar coordinates with radial component $r$ and polar angle $\theta$:

$$x = r\cos\theta$$
$$y = r\sin\theta$$

or the instantiation of the hyperspherical coordinate system for three dimensions, otherwise known as spherical coordinates, with radial component $r$, polar angle $\theta$ and azimuthal angle $\phi$:

$$x = r\cos\theta$$
$$y = r\sin\theta\cos\phi$$
$$z = r\sin\theta\sin\phi$$

More generally, hyperspherical coordinates are composed by a radial coordinate $r$ and $d-1$ angular coordinates $\theta_1, ..., \theta_{d-1}$ where $\theta_1, ...\theta_{d-2}$ are supported over $[0, \pi]$ while $\theta_d-1$ ranges from $[0, 2\pi]$. We outline one common conversion from Cartesian coordinates to hyperspherical coordinates below, and other conversions are equivalent up to permutation of the coordinate axes:

$$x_1 = r\cos(\theta_1)$$
$$x_2 = r\sin(\theta_1)\cos(\theta_2)$$
$$x_3 = r\sin(\theta_1)\sin(\theta_2)\cos(\theta_3)$$
$$\vdots$$
$$x_{d-1} = r\sin(\theta_1)\cdots\sin(\theta_{d-2})\cos(\theta_{d-1})$$
$$x_d = r\sin(\theta_1)\cdots\sin(\theta_{d-2})\sin(\theta_{d-1})$$

Figure 19: Visualization of basis vectors at different points under Cartesian (left) and spherical (right) coordinatate systems. Notice that the Cartesian basis vectors do not change from point-to-point; however, the spherical basis vectors change in both direction and magnitude. Even at the same radius, the $\frac{\partial}{\partial \phi}$ coordinate changes based on the azimuth angle $\theta$ because the same infinitesimal change in $\phi$ will result in a longer (or smaller) change in arclength depending on the radius of the circle at latitude $\theta$.

and the reverse transform from Cartesian coordinates to hyperspherical coordinates:

$$r = \sqrt{(x_1)^2 + (x_2)^2 + ... + (x_d)^2}$$
$$\theta_1 = \arctan 2(\sqrt{(x_d)^2 + ... + (x_2)^2}, x_1)$$
$$\theta_2 = \arctan 2(\sqrt{(x_d)^2 + ... + (x_3)^2}, x_2)$$
$$\vdots$$
$$\theta_{d-2} = \arctan 2(\sqrt{(x_d)^2 + (x_{d-1})^2}, x_{d-2})$$
$$\theta_{d-1} = \arctan 2(\sqrt{(x_d)^2}, x_{d-1})$$

where $\arctan 2(x, y)$ returns the angle measurement in radians over the support $(-\pi, \pi]$ between between $x$ and $y$.

Unlike the Cartesian coordinate system, the hyperspherical basis vectors are not identical over the entire space; they change with position. For instance, moving outwards along $r$ will increase the length of $\frac{\partial}{\partial \theta^i}$ as an infinitesimal change in $\theta^i$ will now cover a larger arclength distance—i.e. the line segment traveled by changing the angle $\theta^i$—than that same infinitesimal change with a smaller $r$. This effect is visualized for three dimensions in Figure 19.

At any given point in hyperspherical coordinates $\tilde{p}$, the transformation from Cartesian basis vectors $\frac{\partial}{\partial x^1}, \frac{\partial}{\partial x^2}, ...$ to hyperspherical basis vectors $\frac{\partial}{\partial r}, \frac{\partial}{\partial \theta^1}, ...$ can be computed with the multivariate chain rule:

$$\frac{\partial}{\partial \theta^i} = \sum_{k=1}^{d} \frac{\partial x^k}{\partial \theta^i} \frac{\partial}{\partial x^k}$$

where $\frac{\partial x^i}{\partial \theta^i}$ can be computed from the coordinate transform functions, i.e. $x_1 = r \cos(\theta_1)$. It is typical to express these relationships in a matrix that transforms an arbitrary vector $v$ in Cartesian coordinates

at point $p$ to its counterpart in hyperspherical coordinates $\tilde{v}$ at $\tilde{p}$:

$$\begin{bmatrix} \frac{\partial}{\partial x^1} & \frac{\partial}{\partial x^2} & \cdots & \frac{\partial}{\partial x^d} \end{bmatrix} \underbrace{\begin{bmatrix} \frac{\partial x^1}{\partial r} & \frac{\partial x^1}{\partial \theta^1} & \cdots & \frac{\partial x^1}{\partial \theta^{d-1}} \\ \frac{\partial x^2}{\partial r} & \frac{\partial x^2}{\partial \theta^1} & \cdots & \frac{\partial x^2}{\partial \theta^{d-1}} \\ \vdots & \vdots & \ddots & \vdots \\ \frac{\partial x^d}{\partial r} & \frac{\partial x^d}{\partial \theta^1} & \cdots & \frac{\partial x^d}{\partial \theta^{d-1}} \end{bmatrix}}_{\text{The Jacobian } J} = \begin{bmatrix} \frac{\partial}{\partial r} & \frac{\partial}{\partial \theta^1} & \cdots & \frac{\partial}{\partial \theta^{d-1}} \end{bmatrix}$$

As illustrated in Figure 19, $J$ does not necessarily have determinant equal to one and changes as a function of position, so the norms of the basis vectors spanning the hyperspherical tangent space change based on position. More generally, this notion of distance is captured by the line element: the length of a line segment resulting from an infinitesimal change along the coordinate axes. The Cartesian line element is given by:

$$ds^2 = (dx^1)^2 + (dx^2)^2 + ... + (dx^d)^2$$

while the hyperspherical line element is:

$$ds^2 = dr^2 + r^2(d\theta^1)^2 + r^2 \sin^2 \theta_1 (d\theta^1)^2 + r^2 \left[ \prod_{i=2}^{d-1} \sin^2 \theta_i \right] (d\theta^{d-1})^2$$

which reflects that distance traveled by small changes in the hyperspherical coordinates "increases" with increasing radius and "decreases" with distance from the equator. To ensure that the norm of the basis vectors does not change during conversion, it is common to renormalize hyperspherical basis vectors to have unit norm for all points. However, a notion of norm is not defined *a priori* for hyperspherical vectors; the metric tensor imposed on this space defines the inner product which in turn defines a sense of arclength.

Using the induced metric from Cartesian coordinates, we can inherit the inner product from Cartesian coordinates on the hyperspherical coordinate system by expressing hyperspherical basis vectors as a linear combination of Cartesian basis vectors and then computing the norm of this resulting vector in the Cartesian tangent space:

$$\left\| \frac{\partial}{\partial \theta^i} \right\| = \sqrt{\left\langle \frac{\partial}{\partial \theta^i} \,,\, \frac{\partial}{\partial \theta^i} \right\rangle}$$

$$= \sqrt{\left[ \sum_{k=1}^{d} \frac{\partial x^k}{\partial \theta^i} \frac{\partial}{\partial x^k} \right] \cdot \left[ \sum_{j=1}^{d} \frac{\partial x^j}{\partial \theta^i} \frac{\partial}{\partial x^j} \right]}$$

$$= \sqrt{\sum_{k=1}^{d} \frac{\partial x^k}{\partial \theta^i} \frac{\partial x^k}{\partial \theta^i} \left[ \frac{\partial}{\partial x^k} \cdot \frac{\partial}{\partial x^k} \right]}$$

$$= \sqrt{\sum_{k=1}^{d} \left( \frac{\partial x^k}{\partial \theta^i} \right)^2}$$

The first fundamental form gives us the normalization constants:

$$\mathcal{I} = \begin{bmatrix} 1^2 & 0 & 0 & \ldots & 0 \\ 0 & r^2 & 0 & \ldots & 0 \\ 0 & 0 & r^2 \sin^2 \theta_1 & \ldots & 0 \\ \vdots & \vdots & \vdots & \ddots & \vdots \\ 0 & 0 & 0 & \ldots & r^2 \prod_{i=1}^{d-1} sin^2 \theta_i \end{bmatrix}$$

as the diagonal represents the inner product $\left\langle \frac{\partial}{\partial \theta^i} \,,\, \frac{\partial}{\partial \theta^i} \right\rangle$, and we would like to renormalize each basis vector to have unit norm: $\left\| \frac{\partial}{\partial \theta^i} \right\| = \sqrt{\left\langle \frac{\partial}{\partial \theta^i} \,,\, \frac{\partial}{\partial \theta^i} \right\rangle}$. Therefore, our normalized hyperspherical basis vectors $\frac{\partial}{\partial \hat{r}}, \frac{\partial}{\partial \hat{\theta_1}}, ...$ become:

$$\frac{\partial}{\partial \hat{r}} = \frac{\partial}{\partial r}$$

$$\frac{\partial}{\partial \hat{\theta}_i} = (\mathcal{I}_{ii})^{-\frac{1}{2}} \frac{\partial}{\partial \theta_i}$$

Using our convention from earlier, we can now compute the transformation from Cartesian basis vectors to normalized hyperspherical basis vectors:

$$\frac{\partial}{\partial \hat{\theta}^i} = (\mathcal{I}_{ii})^{-\frac{1}{2}} \sum_{k=1}^{d} \frac{\partial x^k}{\partial \theta^i} \frac{\partial}{\partial x^k}$$

to compose the normalized "Jacobian" $\hat{J}$:

$$\begin{bmatrix} \frac{\partial}{\partial x^1} & \frac{\partial}{\partial x^2} & \cdots & \frac{\partial}{\partial x^d} \end{bmatrix} \underbrace{\begin{bmatrix} \frac{\partial x^1}{\partial \hat{r}} & \frac{\partial x^1}{\partial \hat{\theta}_1} & \cdots & \frac{\partial x^1}{\partial \hat{\theta}_{d-1}} \\ \frac{\partial x^2}{\partial \hat{r}} & \frac{\partial x^2}{\partial \hat{\theta}_1} & \cdots & \frac{\partial x^2}{\partial \hat{\theta}_{d-1}} \\ \vdots & \vdots & \ddots & \vdots \\ \frac{\partial x^d}{\partial \hat{r}} & \frac{\partial x^d}{\partial \hat{\theta}_1} & \cdots & \frac{\partial x^d}{\partial \hat{\theta}_{d-1}} \end{bmatrix}}_{\hat{J} \in SO(d)} = \begin{bmatrix} \frac{\partial}{\partial \hat{r}} & \frac{\partial}{\partial \hat{\theta}^1} & \cdots & \frac{\partial}{\partial \hat{\theta}^{d-1}} \end{bmatrix} \quad (1)$$

Rescaling the hyperspherical basis vectors to have unit norm at all points causes the matrix $J$ to become the orthogonal matrix with determinant equal to one $\hat{J}$. This set of $d \times d$ matrices belongs to the group $SO(d)$, which represents the set of $d$-dimensional rotations about the origin. Similarly, the backwards change-of-basis $\hat{J}^{-1} = \hat{J}^T$ converts vectors in hyperspherical coordinates to Cartesian coordinates.

As a result, vectors from the tangent space at $p$ in Cartesian coordinates simply rotate to convert to the normalized tangent space at $\tilde{p}$ in hyperspherical coordinates. Specifically, for a point $\tilde{p} = (r, \theta_1, \theta_2, ..., \theta_{d-1})$ and a vector $\tilde{v} = \tilde{c}_1 r + \tilde{c}_2 \theta^1 + ... + \tilde{c}_d \theta^{d-1}$, converting $v = c_1 x^1 + ... + c_d x^d$ from Cartesian to hyperspherical coordinates is the transformation:

$$\tilde{v} = \hat{J}^T v$$

where $\hat{J}$ operates on vector $v$—i.e. $\hat{J}v$—by first rotating by angle $\tilde{c}_2$ in the $x^1 - x^2$ plane (i.e. the $\theta^1$ axis of rotation), then by angle $\tilde{c}_3$ in the $x^2 - x^3$ plane (i.e. the $\theta^2$ axis of rotation), so on and so forth until a final rotation by angle $\tilde{c}_d$ in the $x^{d-1} - x^d$ plane (i.e. the $\theta^{d-1}$ axis of rotation). Composing these rotations together leads to a rotation from $\tilde{p}_0 = (1, 0, 0, ..., 0)$ to $\tilde{p}$:

$$\hat{J}v = (R_{\tilde{p}_0 \to \tilde{p}})v = (R_{\theta_d}^{x^{d-1}-x^d} \cdots R_{\theta_2}^{x^2-x^3} R_{\theta_1}^{x^1-x^2})v$$

$$\hat{J}^{-1}v = \hat{J}^T v = (R_{\tilde{p}_0 \to \tilde{p}})^T v = (R_{\theta_d}^{x^{d-1}-x^d} \cdots R_{\theta_2}^{x^2-x^3} R_{\theta_1}^{x^1-x^2})^T v = R_{\tilde{p} \to \tilde{p}_0} v$$

where we define $R_{\tilde{a} \to \tilde{b}}$ to be the rotation from $\tilde{a}$ to $\tilde{b}$ as described above and $R_{\theta_i}^{x^i-x^j}$ to be the rotation by angle $\theta_i$ in the $x^i - x^j$ plane. Important for our later discussion on the rotation trick, this rotational characteristic causes moving a fixed vector along a curve in hyperspherical coordinates to rotate in Cartesian coordinates.

**Remark 4.** *Using the renormalized transformation in Equation* (1)*, a constant vector field $\tilde{v}$ in hyperspherical coordinates corresponds to a rotated vector field in Cartesian coordinates.*

*Proof.* At Cartesian point $p$ and corresponding hyperspherical point $\tilde{p}$:

$$v_p^T \left[ R_{\theta_d}^{x^{d-1}-x^d} \cdots R_{\theta_2}^{x^2-x^3} R_{\theta_1}^{x^1-x^2} \right] = \tilde{v}_{\tilde{p}}^T$$

$$\left[ R_{\theta_d}^{x^{d-1}-x^d} \cdots R_{\theta_2}^{x^2-x^3} R_{\theta_1}^{x^1-x^2} \right]^T v_p = \tilde{v}_{\tilde{p}}$$

$$[R_{\tilde{p} \to \tilde{p}_0}] v_p = \tilde{v}_{\tilde{p}}$$

so a constant vector field $\tilde{v}$ in hyperspherical coordinates will correspond to a cartesian vector field where each vector at point $p$ is rotated by the rotation that alights $\tilde{p}$ to $\tilde{p}_0$. $\square$

Another important characteristic relates to the metric tensor with normalized hyperspherical basis vectors. We can explicitly compute the induced metric in hyperspherical coordiantes in terms of our renormalized basis vectors:

$$
\begin{aligned}
\hat{\mathcal{I}} &= \begin{bmatrix}
\frac{\partial}{\partial \hat{r}} \cdot \frac{\partial}{\partial \hat{r}} & 0 & 0 & \dots & 0 \\
0 & \frac{\partial}{\partial \hat{\theta}_1} \cdot \frac{\partial}{\partial \hat{\theta}_1} & 0 & \dots & 0 \\
0 & 0 & \frac{\partial}{\partial \hat{\theta}_2} \cdot \frac{\partial}{\partial \hat{\theta}_2} & \dots & 0 \\
\vdots & \vdots & \vdots & \ddots & \vdots \\
0 & 0 & 0 & \dots & \frac{\partial}{\partial \hat{\theta}_{d-1}} \cdot \frac{\partial}{\partial \hat{\theta}_{d-1}}
\end{bmatrix} \\
&= \begin{bmatrix}
(\mathcal{I}_{11})^{-1} \frac{\partial}{\partial r} \cdot \frac{\partial}{\partial r} & 0 & 0 & \dots & 0 \\
0 & (\mathcal{I}_{22})^{-1} \frac{\partial}{\partial \theta_1} \cdot \frac{\partial}{\partial \theta_1} & 0 & \dots & 0 \\
0 & 0 & (\mathcal{I}_{33})^{-1} \frac{\partial}{\partial \theta_2} \cdot \frac{\partial}{\partial \theta_2} & \dots & 0 \\
\vdots & \vdots & \vdots & \ddots & \vdots \\
0 & 0 & 0 & \dots & (\mathcal{I}_{dd})^{-1} \frac{\partial}{\partial \theta_{d-1}} \cdot \frac{\partial}{\partial \theta_{d-1}}
\end{bmatrix} \\
&= \begin{bmatrix}
(\mathcal{I}_{11})^{-1}(\mathcal{I}_{11}) & 0 & 0 & \dots & 0 \\
0 & (\mathcal{I}_{22})^{-1}(\mathcal{I}_{22}) & 0 & \dots & 0 \\
0 & 0 & (\mathcal{I}_{33})^{-1}(\mathcal{I}_{33}) & \dots & 0 \\
\vdots & \vdots & \vdots & \ddots & \vdots \\
0 & 0 & 0 & \dots & (\mathcal{I}_{dd})^{-1}(\mathcal{I}_{dd})
\end{bmatrix} \\
&= \begin{bmatrix}
1 & 0 & 0 & \dots & 0 \\
0 & 1 & 0 & \dots & 0 \\
0 & 0 & 1 & \dots & 0 \\
\vdots & \vdots & \vdots & \ddots & \vdots \\
0 & 0 & 0 & \dots & 1
\end{bmatrix}
\end{aligned}
\tag{2}
$$

which yields the identity matrix. This is perhaps unsurprising: we normalize basis vectors so that $\frac{\partial}{\partial \hat{\theta}^i} \cdot \frac{\partial}{\partial \hat{\theta}^i} = 1$.

Another way to view the renormalized tangent plane transformation is as a change-of-basis in the Cartesian coordiante system, with the basis vectors spanning each tangent space in Cartesian coordinates rotated to align with the directions of hyperspherical basis vectors. Rotating the tangent space at each point in Cartesian coordinates does not change the Euclidean metric tensor—$R^T I R = I$—so it remains the identity. These two formulations are equivalent: renormalizing the basis vectors in the hyperspherical tangent space to have unit norm corresponds to rotating the basis vectors in the Cartesian coordinate system to align with the hyperspherical basis vectors at all points.

### A.12.2 STE AS PARALLEL TRANSPORT

From the description of the STE in Bengio et al. (2013), a gradient vector $\nabla_q \mathcal{L}$ is transported from $q$ to $e$ during the backwards pass in such a way that its direction and magnitude is preserved. Critically, the curve along which $\nabla_q \mathcal{L}$ is transported is not specified; the effect is to simply "copy-and-paste" the vector from $q$ to $e$.

To use the machinery of calculus, we assume that $\nabla_q \mathcal{L}$ is transported from $q$ to $e$ along **any** smooth curve $\gamma(t)$ running from $q$ to $e$. Along this curve, we define the transport of $\nabla_q \mathcal{L}$ at position $\gamma(t)$ simply as $\nabla_q \mathcal{L}$ to emulate how the STE would move $\nabla_q \mathcal{L}$ from $q$ to $\gamma(t)$. Therefore, the direction and magnitude of $\nabla_q \mathcal{L}$ does not change along the curve $\gamma(t)$. An example of this transport is visualized in Figure 20, and in Remark 5, we show this formulation is equivalent to the parallel transport of $\nabla_q \mathcal{L}$ along any curve $\gamma(t)$ from $q$ to $e$ with the Levi-Civita connection.

**Remark 5.** *The Straight Through Estimator (STE) is equivalent to the parallel transport of $\nabla_q \mathcal{L}$ along any curve connecting $q$ to $e$ with the identity metric tensor in Cartesian coordinates using the Levi-Civita connection.*

*Proof.* A vector field $v$ is parallel transported along a curve $\gamma(t)$ if the covariant derivative of $v$ in the direction of $\dot{\gamma}(t)$ is zero. Informally, the change in the vector field $v$ must exactly match how the

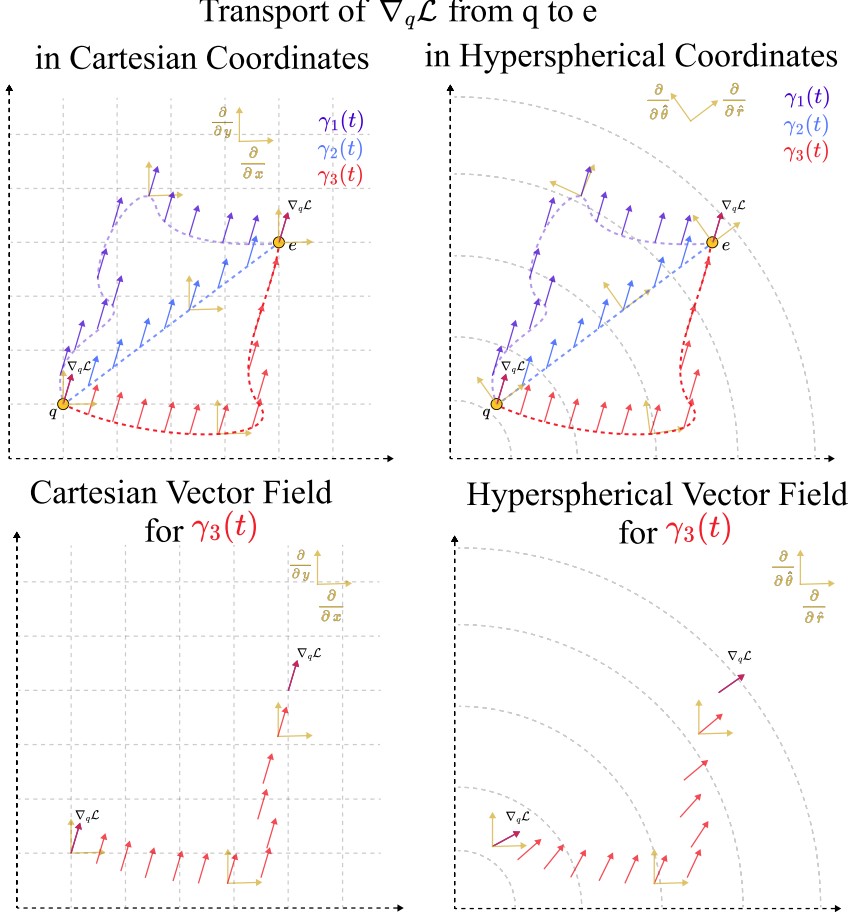

Figure 20: (top) Visualization of vector transport in Cartesian coordinates and renormalized hyperspherical coordinates along curves $\gamma_1(t)$, $\gamma_2(t)$ and $\gamma_3(t)$. Notice the hyperspherical basis changes from point to point. (bottom) Depiction of the transported vector in terms of the basis vectors $\frac{\partial}{\partial x}$ and $\frac{\partial}{\partial y}$ for Cartesian coordinates and $\frac{\partial}{\partial \hat{r}}$ and $\frac{\partial}{\partial \hat{\theta}}$ for hyperspherical coordinates. Notice how the components of $\frac{\partial}{\partial \hat{r}}$ and $\frac{\partial}{\partial \hat{\theta}}$ change for a constant vector field in the Cartesian tangent space.

basis vectors of the tangent plane change along $\gamma(t)$ to remain "parallel" along the curve:

$$\underbrace{\nabla_{\dot{\gamma}(t)} v = \vec{0}}_{\text{Parallel Transport Condition}}$$

Using the identity metric tensor:

$$g_{ij} = \delta_{ij} = \begin{cases} 0 \text{ if } i \neq j \\ 1 \text{ if } i = j \end{cases}$$

with the Levi-Civita connection will result in all zero Christoffel symbols:

$$\Gamma_{ij}^m = \frac{1}{2} g^{mk} \left( \frac{\partial g_{jk}}{\partial x^i} + \frac{\partial g_{ik}}{\partial x^j} - \frac{\partial g_{ij}}{\partial x^k} \right) = 0$$

where $g^{mk}$ is the $m, k$ entry of inverse metric tensor. Computing the covariant derivative for a general curve $\gamma(t)$:

$$\nabla_{\dot{\gamma}(t)} v = \nabla_{\dot{\gamma}_1 e_1 + \dot{\gamma}_2 e_2 + ... + \dot{\gamma}_d e_d} v$$

$$= \sum_{i=1}^{d} \dot{\gamma}_i \nabla_{e_i} v$$

$$= \sum_{i=1}^{d} \dot{\gamma}_i \frac{\partial}{\partial x^i}(v)$$

$$= \sum_{i=1}^{d} \dot{\gamma}_i \underbrace{\frac{\partial}{\partial x^i}(v_1 e_1 + v_2 e_2 + ... + v_d e_d)}_{\text{must be equal to 0 for parallel transport}}$$

Considering the $i^{th}$ term in this summation:

$$0 = (\nabla_{\dot{\gamma}(t)} v)^i = \dot{\gamma}_i \frac{\partial}{\partial x^i}(v_1 e_1 + v_2 e_2 + ... + v_d e_d)$$

$$= \dot{\gamma}_i \frac{\partial}{\partial x^i} \left[ v^k e_k \right]$$

$$= \dot{\gamma}_i \left[ \frac{\partial v^k}{\partial x^i} e_k + v^k \frac{\partial e_k}{\partial x^i} \right]$$

$$= \dot{\gamma}_i \left[ \frac{\partial v^k}{\partial x^i} e_k + v^k \Gamma_{ik}^m e_m \right]$$

$$= \dot{\gamma}_i \left[ \frac{\partial v^k}{\partial x^i} e_k \right]$$

For this equation to hold for an arbitrary $\gamma(t)$, $\frac{\partial v_k}{\partial x^i} = 0$ for $1 \leq k, i \leq d$. Therefore, $v_k$ must be a constant, and vector fields along curves must be constant to satisfy the parallel transport criteria.

Pulling this back to the STE, holding $\nabla_q \mathcal{L}$ constant along the curve $\gamma(t)$ from $q$ to $e$ results in a constant vector field along $\gamma(t)$. The covariant derivative of this vector field is zero, and therefore the STE parallel transports $\nabla_q \mathcal{L}$ from $q$ to $e$. $\qquad \square$

### A.12.3 THE ROTATION TRICK AS PARALLEL TRANSPORT

In this section, we analyze the rotation trick through the lens of geometry. As in Appendix A.12.2, we extend the rotation trick to any smooth curve $\gamma(t)$ connecting $q$ to $e$ and define the transport of $\nabla_q \mathcal{L}$ at $\gamma(t)$ as the rotation trick applied to move $\nabla_q \mathcal{L}$ from $q$ to $\gamma(t)$. This definition allows us to use the structure of calculus, without imposing any prohibitive restrictions on the path taken from $q$ to $e$.

To build visual intuition, Figure 21 illustrates how the rotation trick transforms an initial vector along three different curves $\gamma_1, \gamma_2, \gamma_3$ in both Cartesian coordinates and hyperspherical coordinates with normalized basis vectors. In Cartesian coordinates, the rotation trick changes the components of the basis vectors during transport to follow a rotation; however in normalized hyperspherical coordinates, the components of this vector during transport are constant because the basis vectors themselves rotate.

**Remark 6.** *The rotation trick is equivalent to the parallel transport of $\nabla_q \mathcal{L}$ along any curve connecting $q$ to $e$ with the induced metric in hyperspherical coordinates with the normalized transformation described in Equation* (1) *using the Levi-Civita connection.*

*Proof.* From Equation (2), the metric tensor in hyperspherical coordinates with normalized basis vectors—equivalently, the cartesian coordinate system with each tangent space rotated to align with the hyperspherical frame at every point—is the identity. Therefore, using the Levi-Civita connection

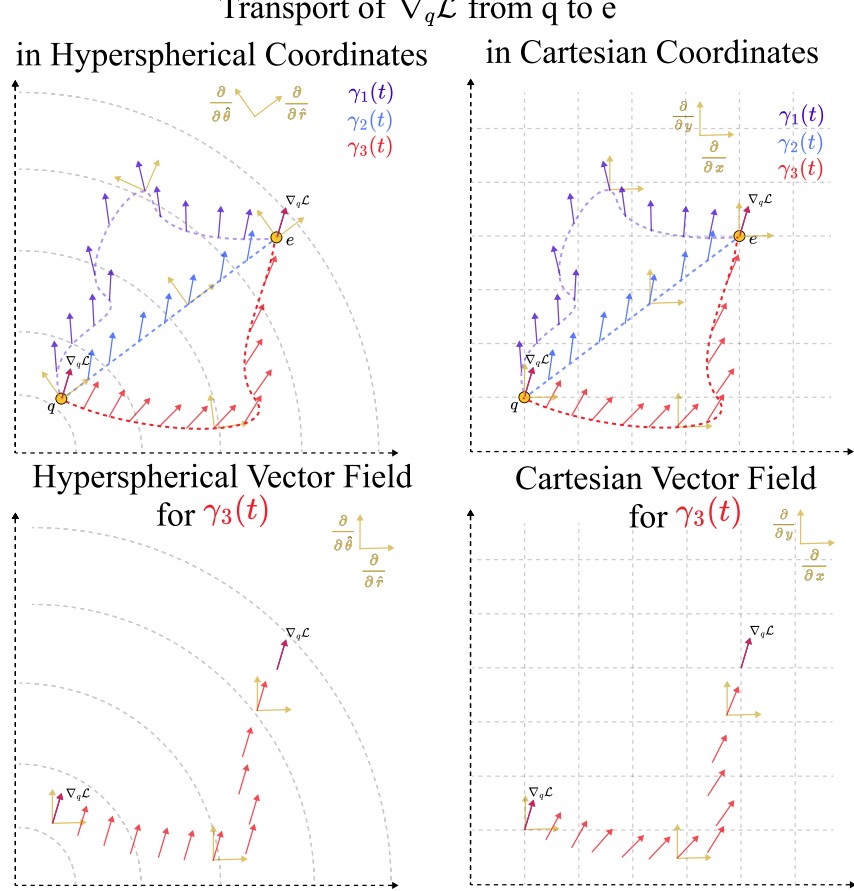

Figure 21: (top) Visualization of vector transport in hyperspherical coordinates with normalized basis vectors and Cartesian coordinates along curves $\gamma_1(t)$, $\gamma_2(t)$ and $\gamma_3(t)$. The vectors along each curve in hyperspherical coordinates *rotate* to stay constant with respect to the natural rotation of the basis vectors. This same rotation in Cartesian coordinates yields a non-constant vector as the Cartesian basis vectors do not change from point to point. (bottom) Depiction of the transported vector in terms of the basis vectors $\frac{\partial}{\partial \hat{r}}$ and $\frac{\partial}{\partial \hat{\theta}}$ for hyperspherical coordinates and $\frac{\partial}{\partial x}$ and $\frac{\partial}{\partial y}$ for Cartesian coordinates. In the former case, the transported vector remains constant with respect to the normalized basis vectors, while in Cartesian coordinates, the components change along $\gamma_3(t)$.

leads to zero Christoffel symbols, and the parallel transport of a vector along any curve keeps the vector constant.

We define $T_pC$ as the tangent space of the Cartesian coordinate system at point $p$ and $T_{\tilde{p}}H$ as the tangent space of the hyperspherical coordinate system with normalized basis vectors at point $\tilde{p}$. It remains to show that for a vector $\nabla_q\mathcal{L} \in T_qC$ and corresponding $\nabla_{\tilde{q}}\tilde{\mathcal{L}} \in T_{\tilde{q}}H$, the transformation of $\nabla_{\tilde{q}}\tilde{\mathcal{L}} \in T_{\tilde{e}}H$ to $T_eC$ will yield $R_{q\to e}\nabla_q\mathcal{L}$ where $R_{q\to e}$ is the rotation trick's transformation, i.e. the rotation that rotates $q$ to $e$.

For a vector $\nabla_{\tilde{q}}\tilde{\mathcal{L}}$ in hyperspherical coordinates at point $\tilde{q} = (1, \theta_1, \theta_2, ..., \theta_{d-1})$ and using the normalized change-of-basis in Equation (1), the corresponding vector $\nabla_q\mathcal{L}$ in Cartesian coordinates is:

$$\nabla_q\mathcal{L}^T = \nabla_{\tilde{q}}\tilde{\mathcal{L}}^T \left[ \hat{J}_{\tilde{q}}^{-1} \right]$$

$$\nabla_q\mathcal{L} = \left[ \hat{J}_{\tilde{q}} \right] \nabla_{\tilde{q}}\tilde{\mathcal{L}}$$

$$= \left[ R_{\tilde{p}_0 \to \tilde{q}} \right] \nabla_{\tilde{q}}\tilde{\mathcal{L}}$$

$$= \left[ R_{\theta_{d-1}} R_{\theta_{d-2}} \cdots R_{\theta_1} \right] \nabla_{\tilde{q}}\tilde{\mathcal{L}}$$

and the corresponding vector $\nabla_{\tilde{q}}\tilde{\mathcal{L}}$ at point $\tilde{e}$ is:

$$\nabla_e \mathcal{L}^T = \nabla_{\tilde{q}}\tilde{\mathcal{L}}^T \left[ \hat{J}_e^{-1} \right]$$

$$\begin{aligned}
\nabla_e \mathcal{L} &= \left[ \hat{J}_e \right] \nabla_{\tilde{q}}\tilde{\mathcal{L}} \\
&= \left[ R_{\tilde{p}_0 \to \tilde{e}} \right] \nabla_{\tilde{q}}\tilde{\mathcal{L}} \\
&= \left[ R_{\tilde{q} \to \tilde{e}} R_{\tilde{p}_0 \to \tilde{q}} \right] \nabla_{\tilde{q}}\tilde{\mathcal{L}} \\
&= R_{\tilde{q} \to \tilde{e}} \left[ R_{\tilde{p}_0 \to \tilde{q}} \nabla_{\tilde{q}}\tilde{\mathcal{L}} \right] \\
&= \left[ R_{\tilde{q} \to \tilde{e}} \right] \nabla_q \mathcal{L}
\end{aligned}$$

which is exactly how the rotation trick transforms the vector. Informally, "copy-and-pasting" the vector $\nabla_{\tilde{q}}\tilde{\mathcal{L}}$ from $\tilde{q}$ to $\tilde{e}$ in hyperspherical coordinates with normalized basis vectors corresponds to rotating $\nabla_q \mathcal{L}$ by the rotation that aligns $q$ to $e$ in Cartesian coordinates. $\square$

In summary, we consider a geometry where the tangent space is spanned by unit norm basis vectors $\frac{\partial}{\partial \hat{r}}, \frac{\partial}{\partial \hat{\theta}^1}, ..., \frac{\partial}{\partial \hat{\theta}^{d-1}}$ that match the direction of the typical hyperspherical basis vectors $\frac{\partial}{\partial r}, \frac{\partial}{\partial \theta^1}, ..., \frac{\partial}{\partial \hat{\theta}^{d-1}}$. The induced metric tensor is the identity, so the parallel transport of a vector along any curve holds its components constant. Converting a vector $\nabla_q \mathcal{L}$ to this tangent space via the normalized transformation in Equation (1), parallel transporting the resulting vector from $\tilde{q}$ to $\tilde{e}$, and then converting it back to Cartesian coordinates corresponds exactly to the rotation trick's transformation.

This is a remarkably simple result; the rotation trick and the STE can be viewed as the same operation. Both parallel transport the gradient $\nabla_q \mathcal{L}$ from $q$ to $e$ in a path-independent manner with the Euclidean metric. The only difference is the coordinate system where parallel transport occurs. The STE employs the Cartesian coordinate system while the rotation trick uses the hyperspherical coordinate system with normalized basis vectors.

