# OpenReview forum: "Restructuring Vector Quantization with the Rotation Trick"
_ICLR.cc/2025/Conference — ICLR 2025 Oral_

### Official Review · Reviewer_DhUX · 2024-10-30

**Soundness:** 4
**Presentation:** 4
**Contribution:** 3
**Rating:** 8
**Confidence:** 4

**Summary:**

This paper proposes a method, reffered to as the "rotation trick", to propagate approximated gradients through the vector quantization layer of VQ-VAEs.
The proposed method addresses the issue of a "straight-through estimator (STE)," which loses information about the positions of encoder outputs relative to codebook vectors.
The capability of the method to increase information capacity and reduce distortion in vector quantization is illustrated with examples and explanation of the "push-pull effect".
The authors conduct experiments across 11 VQ-VAE training settings to demonstrate that the proposed method improves reconstruction performance compared to baseline methods.

**Strengths:**

This paper is well-written and well-motivated. Its review of VQ-VAE, STE, and related work is concise and easy to follow, providing a clear motivation for the study.
The algorithm for the rotation trick is clearly formulated.
The explanation of the effects of the rotation trick, supported by figures, is illustrative.
The effectiveness of the proposed method is investigated through experiments in 11 different settings, which can be considered sufficient.

**Weaknesses:**

Overall, this paper is well-motivated and well-structured, but I would like to raise a point that requires clarification.

The algorithm is designed to preserves the angle between a vector and a gradient, which leads to the push-pull effect described in Section 4.3. However, it appears that using only a rotation, $\frac{\partial \tilde{q}}{\partial e} = R$, satisfies this requirement, and rescaling with $\frac{||q||}{||e||}$ is not necessary for the gradient approximation. (I understand that rescaling is necessary for transforming $e$ into $q$.)
Additionally, it seems that rescaling of gradients may hinder the propagation of gradients when $||q||$ is close to zero.
Providing an explanation for why the rescaling factor $\frac{||q||}{||e||}$ is used for the gradient approximation would clarify the motivation behind this algorithm's design.

**Questions:**

* Why is the rescaling factor $||q|| / ||e||$ used for the gradient approximation? How does it affect training of the encoder and the codebook?
* Minor comment: In Definition 1 (Householder Reflection Matrix), isn't it required that $||a|| = 1$?
* Minor comment: there is a typo: "a a convex combination'" in p3 L6.

---

> ### Author Response · Authors · 2024-11-20
> **Response to Reviewer DhUX Part I**
>
> Thank you for championing our work! Like many lines of research, our path to developing the rotation trick was quite convoluted, and we hoped to convey some of our learnings from the things that didn’t work (Section 3) as well as an understanding of why the rotation trick does work (Section 4). From your review and positive feedback, we’re encouraged that these points came through and were clearly communicated. Specifically, you note that “the paper is well-written and well-motivated”, “the explanation of the effects of the rotation trick is illustrative”, and that our experimental evaluation “can be considered sufficient.”
>
> ### Addressing your questions:
>
> >**Q1: Why is the rescaling factor $||q|| / ||e||$ used for the gradient approximation?**
>
> We were mistaken to not include an explanation of this design choice; both you and Reviewer fmLN raise this point. Afterall, one could use $\tilde{q} = Re + (q-Re)$.detach() instead to transform $e$ into $\tilde{q}$ without scaling the norm of the gradients in the backward pass.
>
> A _desideratum_ of vector quantization is that when $e \approx q$, then $\nabla_e \mathcal{L} \approx \nabla_q \mathcal{L}$ (i.e. when distortion is low, the gradients for both $e$ and $q$ are approximately the same). However when $||e|| \approx 0$ and a Euclidean metric is used to determine the closest codebook vector, the angle between $e$ and $q$ can be obtuse (see Figure 6 of A.1). In this instance, the rotation trick will cause the gradient $\nabla_e \mathcal{L}$ to “over-rotate” and point away from $\nabla_q \mathcal{L}$.
>
> Using a grad scaling of $\frac{||q||}{||e||}$ can fix this. When $||e|| \approx 0$ and $||e|| < ||q||$, the norm of the gradient will be scaled up to push $e$ away from the origin. Pushing $e$ away from the origin makes the angle between $e$ and $q$ more of a factor when computing the Euclidean distance:
> \begin{align*}
>     ||e-q|| &= \sqrt{||e||^2 + ||q||^2 - 2 ||e||||q||\cos \theta}
> \end{align*}
> so $e$ is more likely to map to a different $q$ that forms an acute angle with it as $||e||$ increases.
>
> Now consider if $||q|| \approx 0$ and $||e|| > ||q||$. When this occurs, the update to $e$ will vanish because $\frac{||q||}{||e||} \approx 0$. This behavior may also be desirable because when $q$ is close to the origin, there's a higher likelihood the angle between $e$ and $q$ would be obtuse. $||e||$ remains high and can be mapped to a different codebook vector after the update without moving in a (potentially) bad direction due to an obtuse angle with a very low norm $q$.
>
> We’ve also added an ablation experiment to evaluate this difference in Table 8 and Table 9 of Appendix A.10. Specifically, in the typical VQ-VAE paradigm presented in Table 8, we do not observe a difference between using $\tilde{q} = \frac{||q||}{||e||}Re$ and $\tilde{q} = Re + (q-Re)$. However, for the VQGAN results in Table 9, we find that using the multiplicative gradient scaling factor modestly improves performance. We preview the results from Table 9 below:
>
>
> | Rotation Trick Function            | Latent Shape      | Codebook Size | Codebook Usage (↑) | Quantization Error (↓) | Valid Loss (↓) | r-FID (↓) | r-IS (↑)  |
> |------------------------------------|-------------------|---------------|---------------------|-------------------------|----------------|-----------|-----------|
> | `\|q\|/\|e\|Re`           | `64×64×3`         | `8192`        | 45%                 | 4.0e-4                  | 0.161          | 0.46      | 225.0     |
> | `Re - (q - Re)`                   | `64×64×3`         | `8192`        | 28%                 | 1.5e-3                  | 0.183          | 0.6       | 220.0     |
> | `\|q\|/\|e\|Re`           | `32×32×4`         | `16384`       | 18%                 | 3.3e-4                  | 0.292          | 1.5       | 196.1     |
> | `Re - (q - Re)`                   | `32×32×4`         | `16384`       | 13%                 | 9.4e-4                  | 0.292          | 1.5       | 191.5     |
>
> We also examine other gradient scaling formulations, such as $\tilde{q} = \frac{1}{8||q-e||^2}Re + (q - \frac{1}{8||q-e||^2}Re)$, and discuss possible directions for future research in A.10.2 of the Appendix. To summarize, we believe there is almost certainly a stronger approach to gradient scaling than a fixed $||q||/||e||$ factor throughout training—perhaps by exploring dynamic gradient scaling as is often done in the multi-task learning literature [1,2]---and this is an exciting direction for future work.

---

> > ### Author Response · Authors · 2024-11-21
> > **Response to Reviewer DhUX Part II**
> >
> > >**Q2: Definition 1 correction.**
> >
> > Thank you for pointing this out; we’ve corrected the manuscript.
> >
> > >**Q3: p3 L6 typo.**
> >
> > Another good catch; thanks!
> >
> > ### References:
> > [1] Gradnorm: Gradient normalization for adaptive loss balancing in deep multitask networks, Chen et al., 2018. [ICML]
> >
> > [2] Multi-Task Learning Using Uncertainty to Weigh Losses for Scene Geometry and Semantics, Kendall et al., 2018 [CVPR]
> >
> > ---
> >
> > We would like to thank you again for championing our work – really, it encouraged our entire team. Repeating our remarks to Reviewer fmLN, we’d also like to thank you for prompting us to dig deeper into the effects of gradient scaling. These additional analyses led to a more thorough analysis of the method, adding 2 additional Tables, 2 additional Figures, and several sections to our analysis. Most importantly, it fills in a blank about the approach that was previously missing (i.e., the effect of the norm rescaling).

---

> > > ### Author Response · Authors · 2024-11-24
> > > **Follow-up**
> > >
> > > Dear Reviewer DhUX,
> > >
> > > Thank you again for reviewing our work! As the discussion period ends shortly, we wanted to check if you have any further questions or found our responses helpful?
> > >
> > > Please let us know and thank you for your time!

---

> > > ### Comment · Reviewer_DhUX · 2024-11-25
> > >
> > > Thank you for answering my question about the rescaling factor.
> > > The explanation in Section A.1 and the results in Table 9 are convincing to me.
> > > I do not have any more questions.
> > > I apologize for not having enough confidence to raise my rating to "should be highlighted at the conference", but I have raised the Soundness 3 --> 4.
> > > Thank you again for the extensive investigation.

---

> > > > ### Author Response · Authors · 2024-11-25
> > > >
> > > > Great! We're glad the added analysis answers your question, and we appreciate your engagement during the discussion period!

---

### Official Review · Reviewer_fmLN · 2024-11-03

**Soundness:** 3
**Presentation:** 3
**Contribution:** 3
**Rating:** 8
**Confidence:** 3

**Summary:**

The authors propose an alternative to using straight-through estimation for the gradients of a vector-quantized variational autoencoder (VQ-VAE) to stabilize training and improve codebook usage.

VQ-VAEs are a popular model for data compression and generation, consisting of an encoder and a decoder network. However, instead of passing the encoder output directly to the decoder, the output is first quantized to some codebook vector; these vectors are also learnt during training. However, vector quantization (VQ) is a non-differentiable operation; thus, practitioners usually ignore the quantization operation in the backward pass to enable gradient back-propagation; this is called straight-through estimation (STE). More formally, given some encoder output $e$ and its vector quantized version $q$, STE "sets" the Jacobian $\frac{\partial q}{\partial e} = I$, where $I$ is the identity matrix instead of the all-zero matrix (almost everywhere) as would be dictated by the standard rules of calculus.

In this paper, the authors identify some important shortcomings of using STE in practice, such as training instability and codebook collapse. Based on their observations, they instead suggest "setting" the Jacobian $\frac{\partial q}{\partial e} = \frac{\lVert q \rVert}{\lVert e \rVert} \cdot R$, where $R$ is an orthogonal matrix depending on $e$ such that $q = \frac{\partial q}{\partial e} \times e$.

The authors argue and demonstrate empirically across several experiments that this pick significantly improves codebook utilization, reconstruction loss, and reconstruction quality.

**Strengths:**

The paper's core idea to set the Jacobian to the quantity suggested by the authors is reasonably well-motivated. It can be considered a natural dual to STE, as it uses the multiplicative/polar structure over Euclidean space as opposed to the additive structure used by STE.

The core idea in the paper is neat and simple, and the experimental verification appears reasonably extensive and rigorous.

The paper is well-written, with very nice illustrative figures.

**Weaknesses:**

While I do not think the paper suffers from any major weakness, there are two aspects in which it could be strengthened:
1. The authors do not discuss the limitations of their proposed method. While they present good arguments for when their proposed rotation trick is preferable to STE, have they considered the scenarios in which it might be undesirable?
2. The theoretical motivation for the rotation trick is unclear, other than the fact that it is the natural "multiplicative counterpart" to STE. While I really like the "post hoc" argument that the rotation trick results in a non-constant Jacobian, which might help improve codebook usage, I do not see a good mathematical reason for ensuring that the angles between the vectors and their gradients are preserved, which is better than magnitude and direction. For example, one could argue that an advantage of the identity Jacobian is that it does not shrink or expand space since its determinant is one. This could be solved by combining the STE and the rotation trick by setting (in the authors' notation) $\tilde{q} = R e + (q - R e)$ for an appropriate rotation matrix $R$ and using an appropriate stop-gradient; this would now preserve both magnitude and angle as I understand. In fact, we can even create an entire two-parameter family of gradient approximations by setting (in the authors' notation) $\tilde{q} = \lambda^\alpha R_{\alpha, \beta} e + \beta (q - R_{\alpha, \beta} e)$. Have the authors considered such a solution? If not, could the authors run some small-scale experiments at least to check its performance?

Minor:
 - Fig 2: Increase the tick and axis label font size to match the font size of the caption, as the text is illegible at 100% zoom right now.

**Questions:**

- Suggestion: Figure 2 beautifully illustrates the difference between the original and the STE gradient. To make it even nicer, I think the authors should consider adding the "rotation trick" gradient field as well!
 - Recently, the STE was connected to the deeper theory of numerical integration in [1]. Could the authors comment on whether their method might also have such an interpretation?

## References
[1] Liu, L., Dong, C., Liu, X., Yu, B., & Gao, J. (2024). Bridging discrete and back-propagation: Straight-through and beyond. Advances in Neural Information Processing Systems, 36.

---

> ### Author Response · Authors · 2024-11-20
> **Response to Reviewer fmLN Part I**
>
> We appreciate your positive review of our work! We really like the precision you employ to describe the rotation trick and thoughtfulness in proposing to generalize the algorithm to a larger family of rotation-based gradient estimators. We also *_really_* empathize with your point about searching for an underlying mathematical reason for gradient estimators to non-differentiable functions. This is something we’ve wrestled with quite a bit throughout the research process and hope our explanations below shed some light on how we’ve thought about this.
>
> Moreover, we're encouraged by your thoughtful and positive feedback, particularly your recognition that the method is “neat and simple,” “reasonably well-motivated,” the experimental results are “reasonably extensive and rigorous,” and the manuscript is “well-written, with very nice illustrative figures.”
>
> ### Addressing your questions and concerns below:
>
> >**W1: Limitations of the Rotation Trick.**
>
> One potential limitation is when high numerical precision is required on the quantized vectors during training. As the rotation matrix does not have infinite precision, it will rotate $e$ to $q$ up to a small error term so that the input to the decoder will not be exactly $q$ during training, but rather offset by a very small factor depending on the precision used to compute $R$. Nevertheless, this limitation does not arise during inference as a non-differentiable codebook lookup to transform $e$ into $q$ is used since gradients are not needed during test-time.
>
> A second possible limitation is when either the encoder outputs or codebook vectors are forced to be close to 0 norm (i.e., $||e|| \approx 0$ and/or $||q|| \approx 0$).  In this case, the angle between $e$ and $q$ may be obtuse. When this happens, $\nabla_q \mathcal{L}$ will “over-rotate” the gradient as it is transported from $e$ to $q$ so that $\nabla_q \mathcal{L}$ and $\nabla_e \mathcal{L}$ now point in different directions (i.e. the cosine of the angle between $\nabla_e \mathcal{L}$ and $\nabla_q \mathcal{L}$ will be negative). An example is visualized in Figure 6.
>
> This is undesirable because---when the angle between $e$ and $q$ is obtuse---the rotation trick will violate the assumption that when $e \approx q$, $\nabla_q \mathcal{L} \approx \nabla_e \mathcal{L}$, and it will likely result in worse performance than VQ-VAEs trained with the STE. While obtuse angles between $e$ and $q$ are very unlikely---by design, the codebook vectors should be ``angularly close'' to the vectors that are mapped to them---however, if there is a restriction that forces codewords to have near $0$ norm, then the rotation trick will likely perform worse than the STE.
>
> We appreciate you prompting us to think about the limitations of this approach and have added this discussion to the manuscript in Appendix A.1.
>
> >**W2: Theoretical Motivation and Preserving Gradient Magnitude.**
>
> *Theoretical Motivation is unclear*
>
> Adding a bit of context to how this work developed, we wanted to develop a method that incorporated information from the loss landscape around $q$ into the gradient at $e$. Despite our best efforts for the better part of a month, we couldn’t get this to work, but found the learnings incredibly valuable. These learnings served as the basis for our presentation in Section 3, as well as the foundation of our analyses in Section 4.3, and we hope (and anticipate) they’ll be valuable to other researchers looking at vector quantization or similar problems.
>
> We then started looking at how other fields moved vectors from point a to point b, and took inspiration from parallel transport in differential geometry to develop the rotation trick. As an exercise, imagine a circle with two points $a$ and $b$. Let $\vec{v}$ be a vector tangent to the circle at $a$. How should $\vec{v}$ be moved to point $b$?
>
> There are many ways to answer this. We can translate $\vec{v}$ from $a$ to $b$, thereby preserving its direction and magnitude. Alternatively, we can parallel transport $\vec{v}$ so that its orientation (i.e. tangent to the circle) is preserved. Neither solution is correct per se, but both can be useful in different circumstances (and even become equal to each other in a Euclidean space).
>
> This analogy—along with the post hoc analysis in Section 4.3 and strong empirical results in Section 5—gave us the confidence that this result should be written up, even though we were unable to fix a precise notion of correctness for the vector quantization gradient estimator.

---

> ### Author Response · Authors · 2024-11-20
> **Response to Reviewer fmLN Part II**
>
> >**W2: Theoretical Motivation and Preserving Gradient Magnitude (cont.)**
>
> *Preserving Gradient Magnitude in the Rotation Trick*
>
> This is an excellent point (one that Reviewer DhUX also mentions); we decided to devote a large portion of our compute during the rebuttal period into running large-scale benchmarks to evaluate it. Empirically, we observe that setting $\tilde{q} = Re + (q - Re)$ results in slightly worse performance than $\tilde{q} = \frac{||q||}{||e||}Re$. We summarize these findings in Table 8 and Table 9, and we have also added A.10, A.10.1, and A.10.2 to the Appendix to dig into the differences between these two formulations. We preview the results from Table 9 below:
>
>
> | Rotation Trick Function            | Latent Shape      | Codebook Size | Codebook Usage (↑) | Quantization Error (↓) | Valid Loss (↓) | r-FID (↓) | r-IS (↑)  |
> |------------------------------------|-------------------|---------------|---------------------|-------------------------|----------------|-----------|-----------|
> | `\|q\|/\|e\|Re`           | `64×64×3`         | `8192`        | 45%                 | 4.0e-4                  | 0.161          | 0.46      | 225.0     |
> | `Re - (q - Re)`                   | `64×64×3`         | `8192`        | 28%                 | 1.5e-3                  | 0.183          | 0.6       | 220.0     |
> | `\|q\|/\|e\|Re`           | `32×32×4`         | `16384`       | 18%                 | 3.3e-4                  | 0.292          | 1.5       | 196.1     |
> | `Re - (q - Re)`                   | `32×32×4`         | `16384`       | 13%                 | 9.4e-4                  | 0.292          | 1.5       | 191.5     |
>
>
> One difference is that the gradient scaling term will encourage acute angles between $e$ and $q$ by pushing $e$ away from the origin when $||e|| \approx 0$. We describe this effect in detail in A.10.1 of the Appendix. As discussed in Appendix A.1, obtuse angles cause the rotation trick to “over-rotate” the gradient, so acute angles between $e$ and $q$ are good for the rotation trick.
>
> As you point out, we can consider an entire family of rotation-based gradient approximations by setting $\tilde{q} = \gamma(e) Re +(q - \gamma(e)Re)$. This is a really neat generalization: we create Section A.10.2 of the Appendix to review it in more depth. Specifically, we discuss this generalization, visualize two cases we find especially interesting in Figure 14, and discuss how future work might build on this knowledge to develop better gradient estimators.
>
> >**W3 [minor] Increase tick and axes label font:**
>
> Done!
>
> >**Q1: Reproduce Figure 2 for Rotation Trick:**
>
> This is a good suggestion. We’ve added this visualization in Figure 14.
>
> >**Q2: Connection of the Rotation Trick to Deeper Theory of Numerical Integration.**
>
> This is a very cool question. We took a second, in-depth look at the Reinmax paper. Due to the multiplicative nature of our rotation trick, it is at this point unclear whether, like STE, there exists a connection between the rotation trick and numerical integration. However, it is possible that such a connection may be established in the log-domain where the multiplication can be converted into addition. This parallels the connection between gradient descent and exponentiated gradient (mirror descent) updates. Establishing such connections more rigorously would be a future research direction for us.
>
> ---
>
> We’d like to thank you again for taking the time to provide a thorough and through-provoking review of our work. You prompting us to dig deeper into the effects of gradient scaling led to a more thorough analysis of the rotation trick, adding 2 additional Tables, 2 additional Figures, and several sections to our analysis. Most importantly, it fills in a blank about the method that was previously missing.
>
> If there are any other weaknesses that prevent this submission from being a strong contribution to the conference, please let us know.

---

> > ### Comment · Reviewer_fmLN · 2024-11-22
> >
> > I thank the authors for their detailed rebuttal. I checked the updated manuscript and am mostly happy with it; I have raised my score accordingly.
> >
> > My one issue with the updated manuscript is that much of the appendix is not referenced in the main text: in the main text, I only found references to sections A.6, A.8 and A.11, and a few tables. Especially given that the appendix sections are also very nicely written, I would like to ask the authors to ensure that every section of the appendix is referenced in the main text so that the interested reader is aware of further discussion. It is particularly important that the limitations section is referenced. This section should ideally be in the main text, but I understand that the page limit does not allow for this.

---

> > > ### Author Response · Authors · 2024-11-22
> > >
> > > Thank you for your quick response and engagement during the discussion period!
> > >
> > > We also appreciate this latest suggestion to improve the presentation of our work and ensure that relevant analyses and learnings are not lost in the Appendix. **We commit to ensuring that every section of the Appendix is referenced in the main text, and in particular, referencing or pulling up the Limitations section.** For additional context, we’ve been discussing pushing the ViT-VQGAN and Video Experimental results to the Appendix to pull up excerpts from the Limitations Section and A.6 (Householder Reflection Transformation). We’d also like to add a subsection to Section 4 previewing the analyses in the Appendix and combine Tables 3, 7, and 9. We have held off on making these changes thus far in case other reviewers request additional analyses during the remainder of the review period (that would affect how these sections are organized).
> > >
> > > Thank you again for the quick response, and we’re glad you find the added Appendix sections very nicely written.

---

### Official Review · Reviewer_AkYx · 2024-11-05

**Soundness:** 4
**Presentation:** 3
**Contribution:** 3
**Rating:** 8
**Confidence:** 4

**Summary:**

The authors address the training instability of the straight-through estimator by proposing the rotation trick for vector quantization (VQ), i.e. rotating the gradient $\nabla_q\mathcal{L}$ at the codebook vector $q$ to become the gradient $\nabla_e\mathcal{L}$ at the encoder output $e$ such that the angle between $q$ and $\nabla_q\mathcal{L}$ after the VQ equals the angle between $e$ and $\nabla_e\mathcal{L}$. The rotation trick significantly improves the image reconstruction quality across 11 VQ(-VAE|GAN) training paradigms.

**Strengths:**

1. The rotation trick is novel, intuitive, and mathematically principled
1. Applying two Householder reflectors is computationally efficient, and incurs no additional training cost
1. Strong empirical success across various VQ(-VAE|GAN) training paradigms

**Weaknesses:**

1. The method, i.e. how to achieve the rotation, is not explained clearly in the title or main text. It's probably a good idea to move Appendix A.4 into the main text. In fact, consider renaming this paper to "Restructuring Vector Quantization with the *Reflection* Trick", because the rotation $R$ is really two Householder reflectors chained together (maybe we can get away with one reflection? see below).
1. Appendix A.4 appears to be incorrect: in Remark 3, I think the final reflection/rotation matrix should be $(I-2cc^T)(I-2aa^T)$, or equivalently $(I-2bb^T)(I-2cc^T)$, but not $(I-cc^T)(I-bb^T)$. To make this part easier to understand, perhaps you should re-use the $e, q, r$ notation instead of introducing new symbols $a, b, c$, and include a detailed derivation of why $R = (I-2cc^T)(I-2aa^T) = (I-2bb^T)(I-2cc^T) = I - 2rr^T + 2qe^T$.
1. The effect of the rotation trick is only verified on a high level by codebook usage and quantization error. In particular, Figure 4 is only a diagram ("how we think it works"), not generated from an actual numerical simulation ("how it really works"). Moreover, Figure 4 only shows one gradient update step, and it's unclear how the rotation trick affect the codebook in the long run.
1. The notations are a little confusing. For example, Section 4.3 allows $\tilde{\theta}$ to be negative or exceed $\pi$, but the angle between two vectors is typically defined within $[0, \pi]$. The top panel in Figure 5 also defines $\tilde{\theta}$ in a nonstandard way (when $q$ and $\nabla_q\mathcal{L}$ points at opposite directions, $\tilde{\theta} = 0$ according to the diagram, but normally we say $\tilde{\theta} = \pi$ for this case). In addition, renaming $\tilde{\theta}$ to $\phi$ might avoid some misread.
1. There is an extra "a" on line 113.

**Questions:**

1. What if we just do one Householder reflection, i.e. $R = I - 2rr^T$? It still preserves the angle ($\tilde{\theta} \in [0, \pi]$), but the "push" effect in Section 4.3 becomes "pull", and vice versa. However, as you pointed out, both push and pull are beneficial, so the result should still be good. The mirror symmetry makes me think this will be as effective as $R = I - 2rr^T + 2qe^T$, and it would be great if you could verify this hypothesis experimentally.
1. The rotation/reflection transformation depends on the location of a center, which you implicitly choose to be the origin. Do you need to shift the $q$ and $e$ such that their center/mean is the origin, or does the rotation trick work even if all $q$ and $e$ fall on one side of the origin (e.g. imagine Figure 4, but on a shifted coordinates system $x' = x + 10, y' = y + 10$, such that the coordinate of all points are positive)?

---

> ### Author Response · Authors · 2024-11-20
> **Response to Reviewer AkYx Part I**
>
> This is an excellent review; it’s clear you invested quite a bit of time into thoroughly reading and understanding our work. Your questions about a single reflection and invariance with respect to the origin are exceptional, and we’ve added several pages to the Appendix to address them.
>
> Similarly, we appreciate your positive regard for our work, noting that “the rotation trick is novel, intuitive, and mathematically principled” as well as “the strong empirical success across various VQ(-VAE|GAN) training paradigms”. We address your questions below:
>
> ### Improvements to the Manuscript / Writing / Presentation:
>
> >**W1: Describe how to Compute R in the Main Text.**
>
> As you likely inferred, we moved the discussion of Householder reflections to the Appendix from the main text before the submission due to space limitations. Our thinking was that Householder transformations are not our contribution, and that interested readers could reference Appendix A.6 as suggested in Section 4.2. For the camera ready version, we’ll look into refactoring the paper—perhaps by moving the video results into the appendix—to pull up some of the background on Householder reflections.
>
> >**W2: Remark 3 Incorrect & Detailed Derivation.**
>
> You’re entirely correct; there are multiple mistakes in Remark 3. We sincerely thank you for catching these and have corrected them (see the revised manuscript). We also include a detailed derivation of why $R = I - 2rr^T + 2qe^T$.
>
> >**W3: Figure 4 only a Diagram Not a Numerical Simulation.**
>
> We were mistaken not to include a description of how Figure 4 was created. It is in fact a numerical simulation by applying a small gradient update over 25 steps. It’s not a perfect example: the gradient does not change across batches and we do not update the codebook regions; however, it should faithfully highlight the differences between the STE and the rotation trick. We’ve added Appendix A.11.5 to describe this simulation.
>
> As recommended in your review, we have also added an additional synthetic on Himmelblau’s function across 100 training steps staying as faithful as possible to the actual training paradigm used in VQ-VAEs. This simulation reflects what we see experimentally in our (ViT-)VQ(-VAE|GAN) results: the pre-quantized vectors cluster much more tightly around the codebook vectors (lower distortion) and the objective function’s loss is substantially lower. We have added this analysis to the manuscript in Figure 7, Figure 8, and Appendix A.2.
>
>
> >**W4: Figure 5 Angle Axes.**
>
> Another excellent catch, as you point out, Figure 5 is incorrect in the angle showing $\hat{\theta}$. We’ve updated Figure 5, and based on your later suggestion, have also changed $\tilde{\theta}$ to $\phi$.
>
> For your point about the angle between vectors usually being in $[0, \pi]$, this makes sense for an undirected angle. But if you want to distinguish between the angle needed to rotate $q$ to $\nabla_q \mathcal{L}$ and the angle needed to rotate $\nabla_q \mathcal{L}$ to $q$ (i.e. to differentiate between the clockwise and counterclockwise rotation), then you need the full range from $[0, 2\pi]$. Saying $\frac{-\pi}{2} < \phi < \frac{\pi}{2}$ is a precise way to indicate an acute angle, whereas saying $\phi < \frac{\pi}{2}$ may only indicate the angle lies within the first quadrant of the circle along which radians are typically defined.
>
> >**W5: Typo.**
>
> We appreciate you and reviewer DhUX both pointing out this typo. We have fixed it.

---

> ### Author Response · Authors · 2024-11-20
> **Response to Reviewer AkYx Part II**
>
> ### Key Questions / Additional Analyses
>
> >**Q1: The “Reflection Trick”**
>
> This is an excellent insight. We’ve repeated the analysis in Section 4.3 for using a single reflection rather than a rotation to align $e$ to $q$ in the forward pass to create Figure 12, Figure 13, and A.9 of the Appendix. Specifically, we define the Householder reflection $R = (I - 2 \frac{e-q}{||e-q||}(\frac{e-q}{||e-q||})^T)$ to reflect $e$ to $q$ in the hyperplane orthogonal to $\frac{e-q}{||e-q||}$ (where $e$ and $q$ are rescaled to have unit norm when computing R).
>
> We refer you to the updated manuscript for the full analysis, but list the key findings below:
> 1. When $\nabla_q \mathcal{L}$ is tangent to $q$, the behavior is identical to the rotation trick.
> 2. The component of $\nabla_q \mathcal{L}$ that is orthogonal to $q$ becomes reflected (i.e. reversed) when it is transported to $e$. In particular, if the gradient points “left” at $q$, then it will flip to pointing “right” when it moves to $e$. As a result, the update to $e$ can move in the opposite direction of the direction of the gradient, even when the angle between $q$ and $e$ is very small. This is undesirable as we would hope $\nabla_q \mathcal{L} \approx \nabla_e \mathcal{L}$ when $e \approx q$.
> 3. We evaluated this approach on both settings from Table 3 and the Euclidean codebook with latent Shape 64x64x3 and Codebook size of 8192 from Table 1. Both settings exhibited poor convergence using just a reflection rather than a rotation. Specifically, after one epoch, the validation loss was approximately 3x higher than the rotation trick for both 8192 and 16384 codebook VQGANs in Table 3. For the Euclidean codebook model with latent Shape 64x64x3 in Table 1, the validation loss was approximately 2x higher than the rotation trick after 15 epochs.
>
> This turned out to be quite the nuanced finding; thank you for suggesting this analysis.
>
> >**Q2: Scaling and Invariance to the Origin**
>
> Another excellent point that you note is the rotation trick—unlike the STE—is not invariant to the origin. For all of our empirical results, we use the origin of the vector space at the encoder output: we do not apply any translations or renormalizations.
>
> Based on your suggestion, we directly analyze the case when all points are translated to be farther away from the origin (i.e. Figure 4 but with x’ = x+10 and y’ = y+10). We visualize this case in Figure 10 and discuss the general effect of translations in Appendix A.5. Summarizing this section, as points are translated far away from the origin, the effect of the rotation will become diminished, so that the rotation trick converges to the STE as $d \rightarrow \infty$ in $x’ = x+d$ and $y’ = y+d$.
>
> ---
> Again, we truly appreciate your thorough review, helping us identify issues with the presentation, factual mistakes, and posing relevant (and interesting) questions. We hope our changes improve the paper so that it is now a strong contribution to the conference.

---

> > ### Author Response · Authors · 2024-11-24
> > **Follow-up**
> >
> > Dear Reviewer AkYx,
> >
> > Thank you again for reviewing our work! As the discussion period ends shortly, we wanted to check if you have any further questions or found our responses helpful?
> >
> > Please let us know and thank you for your time!

---

> > > ### Comment · Reviewer_AkYx · 2024-11-25
> > >
> > > Dear authors, thanks for your detailed response and update to the manuscript. I'm also glad that you squashed that bug in your software implementation! The presentation is much better now, and most of my concerns have been addressed, so I'm raising my score.
> > >
> > > Finally, I would appreciate it if you could clean up your code base and release it under a permissive license, so that others can build upon your work.

---

> > > > ### Author Response · Authors · 2024-11-25
> > > >
> > > > Thank you! We appreciate your engagement during the discussion period as well as your thorough review — really; you pointed out two subtle mistakes and suggested several notable follow-up analyses. Incorporating your feedback has markedly improved this work.
> > > >
> > > > With respect to your request, **we commit to cleaning up our codebase and releasing it under an open-source (MIT) license.** We will include a link to the github repository at the end of the Abstract for the camera-ready version.

---

### Official Review · Reviewer_x3Va · 2024-11-08

**Soundness:** 4
**Presentation:** 4
**Contribution:** 3
**Rating:** 8
**Confidence:** 4

**Summary:**

This paper proposes an improvement to the STE method for training VQ-VAE. While STE copies the gradient from the quantizer output to input, this paper proposes the rotation trick, which uses a rotated and scaled gradient according to the value of the quantizer input. This enables the gradient to depend on the value of the quantizer input, whereas in STE it is the same for all inputs that get mapped to the same codebook vector. Experimental results demonstrate superior reconstruction performance and codebook usage across a variety of tasks using VQ-VAE methods.

**Strengths:**

- The method is principled, and I think it is a good and simple idea to improve some of the issues of STE
- The paper motivates the issues with STE well, as well as the rotation trick itself. The explanations with figures are helpful to understand the intuition of the method.
- Experimental results are fairly extensive, encompassing many different scenarios where VQ-based models are used, each showing the superiority of the rotation trick over STE.

**Weaknesses:**

- I have some trouble understanding why the Hessian idea is not good. The explanations provided in the paper are that it is similar to a vanilla auto encoder training, with no quantization. However, this seems only applicable to the "exact gradient" idea and not the Hessian idea. It seems the Hessian idea sets the gradient at e to the gradient at q (what STE does), plus the additional curvature term. Whereas the rotation trick multiples the gradient at q by the scaled rotation matrix. So it seems that both capture the location of e.
- Moreover, I wonder if the analysis in section 4.3 could be extended to the Hessian idea. It seems the Hessian idea also gets the benefits that Section 4.3 argues in favor of the rotation trick. There seems to be some disconnect between this and the experimental results where it was shown the Hessian idea does not perform well. If a stronger explanation is provided, that would be helpful to understand.
- While the experimental settings are fairly extensive, they primarily demonstrate how the rotation trick is superior to STE. I think some other baseline methods that improve upon STE are also necessary for comparison. Namely, those mentioned in Section 2, such as Huh et al 2023, Gautam et al, 2023, and Takida et al, 2022. This would provide comparison between the rotation trick and other methods that go beyond STE.

**Questions:**

Please see the weaknesses above.

---

> ### Author Response · Authors · 2024-11-20
> **Response to Reviewer x3VA Part I**
>
> Thank you very much for taking the time to review our work and for sharing your thoughtful questions and concerns. You make an excellent point about adding additional baselines for comparison with stochastic quantization methods, we have added a section on this to the paper (see more below).
>
> Your observations about the Hessian approximation also resonate deeply. As you likely inferred from the structure of the discussion in Section 3, we initially shared your intuition: we believed that estimating the curvature around $q$ and incorporating this information into the gradient at $e$ (referred to in the paper as "the Hessian approximation”) might lead to significant performance improvements for VQ-VAEs. We devoted considerable effort—weeks of synthetic experiments and benchmarking—to test this hypothesis rigorously. However, despite our best efforts, the approach did not yield the results we hoped for.
>
> While this was initially disappointing, it became a pivotal moment in developing our understanding to allow us to produce the algorithm we present today, the rotation trick. We expand on this transition from the Hessian idea to the rotation trick in Appendix A.3 and add Figure 9 to illustrate several instances where the Hessian idea can lead to undesirable behavior. We also touch on these concepts in our response below.
>
> We are also encouraged by your positive reception of our work and noting its strengths. Specifically, we appreciate your remarks that our “method is principled” and a “good and simple idea to improve upon the STE”, that our paper “motivates the issues with the STE well, as well as the rotation trick itself”, and finally that our “experimental results are fairly extensive, encompassing many different scenarios” and show “the superiority of the rotation trick over the STE”.
>
> ### Addressing your specific questions below:
>
> >**“I have some trouble understanding why the Hessian idea is not good. The explanations provided in the paper are that it is similar to a vanilla auto encoder training, with no quantization. However, this seems only applicable to the ‘exact gradient’ idea and not the Hessian idea."**
>
> Interestingly, the Hessian idea actually approximates the exact gradient up to second order terms. By Taylor’s theorem, we can compute the loss at $e$:
>
> \begin{align*}
> 	\mathcal{L}_e &= \mathcal{L}_q + (\nabla_q \mathcal{L})^T(e-q) + \frac{1}{2}(e-q)^T(\nabla^2_q \mathcal{L})(e-q) + \frac{1}{6}(e-q)^T \nabla^3_q \mathcal{L} (e-q,e-q) + ….
> \end{align*}
>
> So the loss computed by the Hessian idea is different from the loss computed by the exact gradients method by $\mathcal{O}(||e-q||^3)$ which is the remainder term from truncating the Taylor series expansion at the second term.
>
> Moreover, since the Hessian idea approximates the auto encoder gradients, we believe that both methods function similarly and are subject to the same limitations. **We added section A.3 to expand on this point: please let us know if you have any additional suggestions that would help further clarify this section.**
>
>
> >**“It seems the Hessian idea sets the gradient at e to the gradient at q (what the STE does), plus the additional curvature term. Whereas the rotation trick multiplies the gradient at q by the scaled rotation matrix. So it seems that both capture the location of e.”**
>
> This is exactly correct; however, the way the Hessian idea transports the gradient is very different from the Rotation Trick.
>
> The Hessian idea approximates the exact gradient at $e$ with a second-order Taylor expansion around $q$ (i.e. by adding the curvature term to the gradient at $q$). It approximates the loss landscape at $e$ and transforms the gradient to be more aligned with the direction of steepest descent. This is similar to the exact gradients idea, that exactly computes the loss landscape at $e$ to find the direction of steepest descent.
>
> To summarize, the Hessian and exact gradient methods move the gradient based on the loss landscape, whereas the rotation trick (and the STE) transport the gradient in a way that is independent of the loss landscape. We have added Appendix A.3 to discuss several instances where using the loss surface to transport $\nabla_q \mathcal{L}$ from $q$ to $e$ can lead to undesirable behavior and visualize these effects in Figure 9.

---

> ### Author Response · Authors · 2024-11-20
> **Response to Reviewer x3VA Part II**
>
> >**"It seems the Hessian Idea also gets the benefits that Section 4.3 argues in favor of the rotation trick. There seems to be some disconnect between this and the experimental results where it was shown that the Hessian idea does not perform well. If a stronger explanation is provided, that would be helpful to understand."**
>
> A desirable property of vector quantization is that when $e \approx q$ (i.e. low distortion), $\nabla_e \mathcal{L} \approx \nabla_q \mathcal{L}$. Informally, if a codeword matches a vector very well, then they should move together because this codeword approximates this vector with very little distortion. In the STE and the rotation trick, this will always be the case; however in exact gradients and the Hessian approximation, it is not guaranteed since it depends on the loss surface in the Voronoi partition. This is undesirable because $q$ can move in one direction (the direction that minimizes the loss for the decoder), but $e$ can move in the opposite direction, even when it is very close to $q$ to begin with (i.e. a “good” fit with low distortion). We added Figure 9 to visualize this case.
>
> Another undesirable effect is that—when using the loss surface to move gradients—even points that are very close to each other within the same Voronoi region can have drastically different gradients. This effect is visualized in Figure 9 of the Appendix, specifically the bottom partition.
>
> We have added an in-depth discussion of this point to A.3 of the Appendix, and as you suggested, reproduced the analysis of Section 4.3 for the Hessian idea in Figure 9.
>
>
> >**[Aside]**
>
> As an aside, [1] finds that estimating the gradient to second order accuracy (i.e. the Hessian approximation) is beneficial for VAEs that use a discrete distribution (rather than the typical isotropic Gaussian). This is a great result, and our findings do not contradict theirs since the desiderata for vector quantization are very different from discrete VAEs.
>
>
> >**Comparison with VQ-VAEs that use Non-Deterministic Quantization.**
>
> We’ve added Table 6, Table 7, and A.4 of the Appendix to include comparisons to VQ-VAEs that use stochastic quantization, specifically focusing on industry-relevant benchmarks (i.e. the VQGANs used in latent diffusion). We preview the results of Tables 6 and 7 below:
>
>
> | Approach                     | Latent Shape      | Codebook Size | Codebook Usage | Quantization Error (↓) | Valid Loss (↓) | r-FID (↓) | r-IS (↑)  |
> |------------------------------|-------------------|---------------|----------------|-------------------------|----------------|-----------|-----------|
> | VQGAN                        | `64×64×3`        | `8192`        | 15%            | 2.5e-3                  | 0.183          | 0.53      | 220.6     |
> | Gumbel VQGAN                 | `64×64×3`        | `8192`        | 4%             | ---                     | 0.197          | 0.60      | 219.7     |
> | VQGAN w/ Rotation Trick      | `64×64×3`        | `8192`        | 86%            | 1.7e-4                  | 0.142          | 0.27      | 228.0     |
> | VQGAN                        | `32×32×4`        | `16384`       | 2%             | 1.2e-2                  | 0.385          | 5.0       | 141.5     |
> | Gumbel VQGAN                 | `32×32×4`        | `16384`       | 12%            | ---                     | 0.3031         | 1.7       | 189.5     |
> | VQGAN w/ Rotation Trick      | `32×32×4`        | `16384`       | 27%            | 2.4e-4                  | 0.269          | 1.1       | 200.2     |
>
> | Approach                  | Codebook Usage (↑) | Rec. Loss (↓) | Quantization Error (↓) | Rec. Loss (↓) | r-FID (↓) | r-IS (↑)  |
> |---------------------------|--------------------|---------------|-------------------------|---------------|-----------|-----------|
> | **Codebook Lookup: Euclidean**, **Latent Shape:** `64×64×3`, **Codebook Size:** `8192`                                                                           |
> | VQ-VAE                   | 100%              | 0.028         | 1.0e-3                 | **0.030**     | 19.0      | 97.3      |
> | Gumbel VQ-VAE            | 39%               | 0.054         | ---                     | 0.058         | 28.6      | 74.9      |
> | VQ-VAE w/ Rotation Trick | 99%               | 0.028         | 1.4e-4                 | **0.030**     | **16.5**  | **106.3** |
>
>
> ### References
> [1] Liu, L., Dong, C., Liu, X., Yu, B., & Gao, J. (2024). Bridging discrete and back-propagation: Straight-through and beyond. Advances in Neural Information Processing Systems, 36.
>
> ---
>
> We sincerely appreciate you pointing out areas where our presentation could be improved and asking us to highlight the differences between the Hessian approximation and the rotation trick gradient estimators. We hope our changes improve the paper so that it is now a strong contribution to the conference.

---

> > ### Author Response · Authors · 2024-11-24
> > **Follow-up**
> >
> > Dear Reviewer x3VA,
> >
> > Thank you again for reviewing our work! As the discussion period ends shortly, we wanted to check if you have any further questions or found our responses helpful?
> >
> > Please let us know and thank you for your time!

---

> > > ### Comment · Reviewer_x3Va · 2024-11-25
> > >
> > > Dear authors,
> > >
> > > Your response has addressed all my concerns. I find the explanations regarding the Hessian/exact gradients idea, along with the figures comparing how their gradients behave versus the rotation trick, extremely helpful, and provide a convincing explanation as to why they may not work in practice. The experiments demonstrating superiority of rotation trick over stochastic quantization are also convincing that the proposed method makes significant progress over other methods attempting to go beyond STE. I personally find the inclusion of these results to have strengthened the paper, and hope the authors agree. I have raised my score.
> > >
> > > This is not necessary for the review period, but I would recommend the authors to integrate A.4 in main text and reference A.3 within sections 3 and 4 of the main text to help the reader.

---

> > > > ### Author Response · Authors · 2024-11-25
> > > >
> > > > Thank you! We appreciate your engagement during the discussion period and wholeheartedly agree that the changes you suggested—namely, going deeper with the Hessian/exact gradient analysis and adding a comparison to stochastic quantization methods—have decidedly strengthened this work.
> > > >
> > > > Addressing your request, **we commit to integrating A.4 into the main text and referencing or pulling up the discussions from A.3 to Section 3 and 4** for the camera-ready version. We will begin working on this reorganization today.
> > > >
> > > > Thank you again for reviewing this work, leaving constructive feedback that improves the paper, and participating in the discussion period!

---

### Author Response · Authors · 2024-11-20
**Meta Response**

We would like to thank each reviewer for considering our work, and their helpful suggestions for improving the clarity and presentation of our research. We are also encouraged by its positive reception, with the reviewers agreeing that our work is:
1. “Mathematically principled and well-motivated” (Reviewers x3Va, AkYx, fmLN, and DhUX)
2. “Extensive, rigorous, and very strong experimental results” (Reviewers x3Va, AkYx, fmLN, and DhUX)
3. “Well-written, easy to understand, illustrative figures” (Reviewers x3VA, fmLN, and DhUX)

Based on their questions and feedback, we have updated the manuscript and added 8 additional pages to the Appendix—changes highlighted in red—with new analyses, experimental results, and clarifications.

### Changes to the manuscript
* Reviewer x3VA’s suggestions:
  * Clarify the Hessian approximation baseline (Figure 9, Appendix A.3)
  * Benchmark stochastic quantization methods (Table 6, Table 7, Appendix A.4)
* Reviewer AkYx’s suggestions:
  * Fix the Householder section (Appendix A.6)
  * Run a numerical simulation comparing STE to rotation trick (Figure 7, Figure 8, Appendix A.2)
  * Improve Section 4.3 notation (Figure 5, Section 4.3)
  * Describe creation of Figure 4 (Appendix A.11.5)
  * Use a single reflection rather than a rotation (Figure 12, Figure 13, Appendix A.9)
  * Evaluate shifting the origin (Figure 11, Figure 10, Appendix A.5)
* Reviewer fmLN’s suggestions:
  * Discuss possible limitations (Figure 6, Appendix A.1)
  * Analysis into gradient scaling by $\frac{||q||}{||e||}$ (Table 8, Table 9, Figure 14, Appendix A.10, Appendix A.10.1, Appendix A.10.2)
  * Add gradient fields for the rotation trick (Figure 14)
* Reviewer DhUX’s suggestions:
  * Analysis into gradient scaling by $\frac{||q||}{||e||}$ (Table 8, Table 9, Figure 14, Appendix A.10, Appendix A.10.1, Appendix A.10.2)

We hope these changes improve the presentation of our work and turn it into a strong contribution to the conference.

### A Note Regarding Table 3 Results
After the initial submission deadline, we found a bug in our implementation of the rotation trick for the experiments in Table 3. Specifically, we divided by $||e||$ twice in the forward pass (so the input to the decoder would be a codebook vector with its norm arbitrarily scaled up or down).

Fixing this bug further improves the performance of the rotation trick. For the 16384 codebook model, codebook utilization improves from 2% to 27%, r-FID from 1.6 to 1.1, and r-IS from 190.3 to 200.2. Similarly, for the 8192 codebook model, codebook utilization improves from 10% to 86%, r-FID from 0.38 to 0.27, and r-IS from 222.4 to 228.0.

---

### Public Comment · ~Xianghong_Fang1 · 2024-11-30
**Reported Quantization Errors in Tables 1 are Unfair and Meaningless.**

Dear Program Chairs, Senior Area Chairs, Area Chairs, All Reviewers, Authors, and All Readers,

I am a PhD student in University of Toronto, and I have four-year's experience in Vector Quantization (VQ). I am highly skeptical about the reported quantization error improvements in Tables 1 and 2. My reservations are based on the following reasons:
- Suppose there are a feature vector distribution $\mathcal{P}_A$ and a code vector distribution $\mathcal{P}_B$. I have observed that the quantization error is minimized when feature vectors and code vectors are identically distributed, i.e., $\mathcal{P}_A=\mathcal{P}_B$.
- The scale of distribution has a significant impact on the quantization error. Suppose feature vectors and code vectors are sampled from $\mathcal{N}(0, \sigma^2 I)$ or $\text{Unif}(-\zeta, \zeta)$, The variation in quantization error lower bound can be significantly large when $\sigma$ varies. We also provide some experimental results as follow (We provide the code in the Github (https://github.com/sunset-clouds/Quantization-error), every one can justify this):
- For Gaussian distribution $\mathcal{N}(0, \sigma^2 I)$ (e-1=0.1),
 | $\sigma$ | 0.0001 | 0.001 | 0.01 | 0.1 | 1.0|
| --- | --- | --- | --- | --- | --- |
| Quantization Error |   1.25e-8  |   1.25e-6  |    1.25e-4  |    1.24e-2   |   1.25   |

- For uniform distribution $\text{Unif}(-\zeta, \zeta)$ (e-1=0.1),
 | $\zeta$ | 0.0001 | 0.001 | 0.01 | 0.1 | 1.0|
| --- | --- | --- | --- | --- | --- |
| Quantization Error |   3.27e-9  |   3.27e-7  |    3.27e-5  |    3.27e-3   |   0.327  |
- The core issue is **how the authors ensure that the feature vectors across different methods are identically distributed**. If the feature vector distributions are not the same, the reported improvements in quantization error are meaningless, as the reported improvements might be **attributable to differences in distribution scales rather than the effectiveness of the incorporation of the Rotation Trick**. For example, by using vanilla  STE in [1], **I can achieve a more significant reduction in quantization error through regularization of the distribution scales** from $\sigma=1$ to $\sigma=0.001$ (quantization error from $1.25$ to $1.25e-6$). Therefore, the authors should provide a clear explanation demonstrating that the quantization error improvement is indeed due to the effectiveness of the incorporation of the Rotation Trick, rather than merely a result of altered distribution scales.
- Based on my experience, the quantization error comparison in image reconstruction experiments is almost impossible. In different vector quantization (VQ) methods, encoders receive entirely different gradient updates, and thus feature vector distribution are totally different.

The authors need to explain **whether the incorporation of the Rotation Trick will not inflence the distribution scales**; otherwise, the quantization error comparisons provided in Tables 1 and 2 are entirely meaningless. If authors fail to address my concern, quantization error comparison should be removed in the Table 1,2,3,4,5.

We kindly ask the area chair, all reviewers, and readers to take note of my comments and request the authors to respond to this, as I have provided solid evidence to support my concerns.

**I request that the authors provide a detailed explanation for the observed improvement in quantization error within the scope of their study, and I have not requested a comparison with other research papers.**

- [1] Neural Discrete Representation Learning, NeurIPS 2017

Best Regards

---

> ### Comment · Reviewer_DhUX · 2024-12-02
>
> Thank you, Xianghong Fang, for bringing this issue to my attention. I appreciate your insight.
>
> I agree that the scale of quantization error $||e-q||^2_2$ highly depends on the scales of $e$ and $q$. To eliminate this effect, reporting quantization errors normalized by the variance of $e$ or $q$ would be more reliable, for example.
>
> Looking at Figure 8, there does not seem to be a drastic change in the scale of either feature vectors or code vectors in this synthetic example. However, addressing this issue in other tables would strengthen the paper's arguments and make it more convincing.
>
> **I would also appreciate it if authors could respond to this issue.
> If it entails "significant experiments", providing comments and explanations would be helpful.**
> On the other hand, the tables demonstrate improvement in codebook usage and reconstruction metrics.
> **Therefore, I maintain my assessment this time.**
>
> Best regards.

---

> ### Author Response · Authors · 2024-12-02
> **Public Response to Xianghong Fang**
>
> Dear Xianghong Fang,
>
> We appreciate your engagement with our work and for bringing [1] to our attention. While the content of your comment has changed substantially over the past two days (8 revisions), we will try our best to address your current concerns in addition to some of your earlier points.
>
> >Effectiveness of the proposed method.
>
> In a prior version of your comment, you question the “actual effectiveness of the proposed method,” and raise concerns over the reporting of quantization error in our results. Quantization error is one of many metrics that we report such as reconstruction loss, validation loss, codebook usage, r-FID, and r-IS. Even then, it is less important than metrics of reconstruction quality like r-FID and r-IS to measure VQ-VAE performance, as those more directly measure the model’s performance on the reconstruction task itself. Given this, even if we were to ignore any improvements in quantization error, our results as a whole paint a clear picture that the rotation trick is effective: it improves each other measure of VQ-VAE performance across 11 different VQ-VAE training settings.
>
> >Reporting of quantization error metric.
>
> Your comment also raises an interesting discussion of quantization error itself, and specifically whether it is still a valid metric to understand vector quantization techniques. In this regard, we make the following points:
> 1. Quantization error is a standard metric in compression literature (see Section 10.2 of *Elements of Information Theory*, Cover & Thomas 1991). Over the past 70 years, it has been applied to compare different compression schemes, including those that encode the input in various ways so that the feature vector distributions are not identical.
> 2. In the context of VQ-VAEs, it is standard practice to record quantization error [3,4,5,6]. Mirroring these conditions in our results allows for meaningful comparisons between our method and previous works in this field. For Table 2 specifically, we use the “quant_error” field from Tensorboard that [3] created.
> 3. In a prior revision of your comment, you extensively referred to [1]. We appreciate you bringing this work to our attention. They make the assumption, as you do, that the feature vector distribution and codebook vector distribution should be identically distributed and that both distributions can be modeled as a Gaussian. In our work, we follow the examples of [3,4,5,6] and do not make these assumptions. After reviewing [1] and the author-reviewer discussion on OpenReview for [1], we agree with the reviewers of [1] that these assumptions are non-standard and potentially flawed. As such, we do not find the evidence compelling enough to justify incorporating these assumptions into our work at this moment. However, if future work builds on the methods proposed by [1] and prove them to be superior to what is established in the field, we will take this into consideration for future publications.
>
> Thank you again for engaging with our work, and please let us know if you have further concerns.
>
> Sincerely,
>
> Submission 11319 Authors
>
> ---
>
> [0] Elements of Information Theory, Cover & Thomas 1991
>
> [1] Vector Quantization By Distribution Matching, ICLR 2025 submission (https://openreview.net/forum?id=nS2DBNydCC&noteId=QblMp22hOC)
>
> [2] SimVQ: Addressing Representation Collapse in Vector Quantized Models with One Linear Layer, ICLR 2025 submission (https://openreview.net/pdf/2135eca9eb79a8115d813f0179152c30dbce9fda.pdf)
>
> [3] Taming Transformers for High-Resolution Image Synthesis, CVPR 2021
>
> [4] High-Resolution Image Synthesis with Latent Diffusion Models, CVPR 2022
>
> [5] Vector-Quantized Image Modeling with Improved VQGAN, ICLR 2022
>
> [6] Neural Discrete Representation Learning, NeurIPS 2017

---

> ### Author Response · Authors · 2024-12-02
> **Response to Reviewer DhUX**
>
> Dear Reviewer DhUX,
>
> We appreciate your engagement! We agree with your assessment that the “tables demonstrate improvement in codebook usage and reconstruction metrics”, and view quantization error as less important than reconstruction metrics like r-FID and r-IS, that are typically used to compare the performance among VQ-VAE models.
>
> Inspired by your comment, we have re-run the Figure 8 simulation (minimization of Himmelblau’s function) under two conditions:
> 1. The feature vectors and codebook vectors are sampled from a higher variance distribution.
> 2. The feature vectors and codebook vectors are shifted to have higher norms with the loss surface appropriately translated.
>
> For the first case, rather than sampling uniformly from [-5,5] along each axis (as is done in Figure 8), we now sample from [-10, 10]. We repeat the simulation, following the procedure described in Section A.2: 100 epochs, codebook loss replaced by an EMA, learning rate of 1e-3, etc. We visualize the results at the anonymous website https://sites.google.com/view/rotation-trick-images. The average quantization errors for both distributions are roughly the same. For the Uniform(-5,5) distribution, the average quantization error is 0.120 and for the Uniform(-10,10) distribution, it is 0.149.
>
> For the second case, we sample both the feature vectors and codebook vectors from the Uniform(5,15) distribution and correspondingly translate Himmelblau’s function by 10 units along each axis:
> $$f(x,y) = ((x-10)^2 + (y-10) - 11)^2 + ((x-10) + (y-10)^2 - 7)^2$$
> We visualize the results at the anonymous website https://sites.google.com/view/rotation-trick-images. As expected, the feature vectors and codebook of the converged model have larger norms than the Uniform(5,5) case; however, the average quantization error remains largely unchanged at 0.123 (difference of 0.003).
>
>  As both simulations follow the training paradigm employed in VQ-VAEs, we believe they may better model VQ-VAE behavior than the toy example presented by Xianghong, which simply samples two sets of points from the same distribution and then computes the minimum distance. A shortcoming with this latter approach is that it does not account for the effects of minimizing the commitment and codebook loss, which pulls feature and codebook vectors together throughout training.
>
> Concretely, according to Xianghong’s model and their released code on github, for 100 feature vectors and 10 codebook vectors (i.e. the settings of Figure 8), we calculate the quantization error for the Uniform(-5,5) distribution to be 4.987 and the quantization error for the Uniform(-10,10) distribution to be 15.681. Our simulation found this difference to be much smaller: 0.120 vs. 0.149 and well within the standard error from changing the random seed of Uniform(-5,5).
>
> Sincerely,
>
> Submission 11319 Authors

---

> ### Public Comment · ~Xianghong_Fang1 · 2024-12-02
> **Discussion with Reviewer DhUX**
>
> Dear Reviewer DhUX,
>
> Thanks for your discussion! The distribution scale actually has a significant impact on the quantization error. The changes may not appear drastic in the Figure 8 because we visualize the logarithm of the quantization error.
>
> - For Gaussian distribution $\mathcal{N}(0, \sigma^2 I)$ (e-1=0.1),
>  | $\sigma$ | 0.0001 | 0.001 | 0.01 | 0.1 | 1.0|
> | --- | --- | --- | --- | --- | --- |
> | Quantization Error |   1.25e-8  |   1.25e-6  |    1.25e-4  |    1.24e-2   |   1.25   |
>
> - For uniform distribution $\text{Unif}(-\zeta, \zeta)$ (e-1=0.1),
>  | $\zeta$ | 0.0001 | 0.001 | 0.01 | 0.1 | 1.0|
> | --- | --- | --- | --- | --- | --- |
> | Quantization Error |   3.27e-9  |   3.27e-7  |    3.27e-5  |    3.27e-3   |   0.327  |
>
> I encourage the authors to provide a **quantization error improvement in a fair and atomic setting**, for example, by setting the feature distribution to a fixed Gaussian or Uniform distribution in the camera-ready version.
>
> Best Regards,
> Xianghong Fang

---

> ### Public Comment · ~Xianghong_Fang1 · 2024-12-02
> **Discussion with Authors**
>
> Dear Authors,
>
> Thanks for your clarification! I encourage the authors to run cases where feature vectors are sampled from Unif(1.0, 1.0), Unif(-0.1, 0.1), Unif(-0.01, 0.01), and Unif(-0.001, 0.001), **the quantization error would exhibit an order-of-magnitude reduction**.
>
> - For Gaussian distribution $\mathcal{N}(0, \sigma^2 I)$,
>  | $\sigma$ | 0.0001 | 0.001 | 0.01 | 0.1 | 1.0|
> | --- | --- | --- | --- | --- | --- |
> | Quantization Error |   1.25e-8  |   1.25e-6  |    1.25e-4  |    1.24e-2   |   1.25   |
>
> - For uniform distribution $\text{Unif}(-\zeta, \zeta)$,
>  | $\zeta$ | 0.0001 | 0.001 | 0.01 | 0.1 | 1.0|
> | --- | --- | --- | --- | --- | --- |
> | Quantization Error |   3.27e-9  |   3.27e-7  |    3.27e-5  |    3.27e-3   |   0.327  |
>
> It is quite normal that the quantization error remains unchanged between Unif(-5, 5) and Unif(5, 15), as demonstrated in Figure 8, the distribution mean has a minimal impact on the quantization error.
>
> Notably, **[3][4][5][6] listed by the authors have not reported quantization error comparison in the image reconstruction experiments**. This is because it cannot ensure a fair setting as I previously claimed. I encourage the authors to provide a quantization error improvement in a fair and atomic setting, for example, by setting the feature distribution to a fixed Gaussian or Uniform distribution (not based on [1]) in the camera-ready version. I just hope the authors could delete the quantization error comparison in Table 1,2,3,4,5, and **report the quantization error improvement in another table under a fair setting**.
>
> Best Regards,
> Xianghong Fang

---

> ### Comment · Reviewer_AkYx · 2024-12-02
>
> Dear All,
>
> I agree with the authors and reviewer DhUX that the quantization error is only an auxiliary metric, and it's the other end-to-end metrics reconstruction metrics that really demonstrated the effectiveness of this paper. However, since Xianghong is really unhappy about this metric, let's discuss this in detail.
>
> From my understanding, Xianghong's concern is two-fold: (i) the scaling of $q$, and (ii) the normalized distribution of $q$. For (i), the scaling can probably be addressed by reporting the signal-to-noise ratio $\frac{\operatorname*{E}\left[\lVert e - q \rVert_2^2\right]}{\operatorname*{E}\left[\lVert e \rVert_2^2\right]}$ in add to the quantization error $\operatorname*{E}\left[\lVert e - q \rVert_2^2\right]$. For (ii), I have a few comments:
> 1. I believe the authors have already fulfilled the request to set the feature distribution to a fixed Gaussian/Uniform distribution, since both the codewords $q$ and pre-quantized vectors $e$ in Figure 8 are initialized from a uniform distribution on $[-5, 5]^2$.
>     1. The authors should probably specify the initialization distribution as $\operatorname*{Unif}[-5, 5]^2$ instead of just saying "Points for both the STE and the rotation trick simulation use the same random initialization for both codewords and pre-quantized vectors," but this is not a big deal.
> 2. I think the purpose of the rotation trick is to make the feature distribution more evenly distributed across the Voronoi regions, and thus achieve a higher codebook utilization. In particular, the authors apply gradient updates to the encoder such that the pre-quantized vectors $e$ have an easy-to-quantize mixture distribution (as illustrated in Figure 8). Forcing the feature distribution to Gaussian/Uniform defeats the whole point.
> 3. I'm not sure if the suggested experiment is even feasible: we must update the encoder with the gradients, but simultaneously forcing it to produce Gaussian/Uniform distributed pre-quantized vectors $e$. How would you enforce that?
>
>
> Bests.

---

> ### Public Comment · ~Xianghong_Fang1 · 2024-12-03
> **Discussion to Reviewer AkYx**
>
> Dear Reviewer AkYx,
>
> Thanks for your discussion. I believe the reviewer have misunderstood my concerns. Specifically, my concerns pertain to the validity of the quantization error improvements reported in Tables 1, 2, 3, 4, and 5. **The improvement in quantization error is likely not a result of the incorporation of the rotation trick, but rather due to changes in the distribution scales**. Under the VQ-VAE setting, the quantization error can vary significantly with different random seeds, and these variations are often influenced by the alterations in distribution scales.
>
> The authors may choose not to report quantization error (the practice of exitsing VQ papers), but if they do, **it is crucial that authors cannot claim the quantization improvement is due to the rotation trick unless the results are obtained under a fair and controlled setting**.  This would require the experimental setup to carefully control for any changes in distribution scales, ensuring that the observed improvements are solely attributable to the rotation trick and not other confounding factors. In an unfair setting, **such reported improvement is misleading**. **If the authors cannot prove that the quantization error improvement is due to the use of the rotation trick, then reporting the quantization error itself is meaningless.** Therefore, I strongly encourage the author to remove the quantization error metric in their experimental results.
>
> I do not require the authors to conduct the quantization error improvement experiments under a fair setting, as finding such a setting is indeed challenging. Instead, I suggest that the authors do not report the quantization metric in their paper.
>
> Best Regards,
> Xianghong Fang

---

### Meta-Review · Area_Chair_GYsF · 2024-12-20

**Metareview:**

All reviewers appreciate the paper’s contribution and strongly recommend accepting the paper. The AC concurs, after reading the paper quickly.

**Additional Comments On Reviewer Discussion:**

NA

---

### Decision · Program_Chairs · 2025-01-22

Accept (Oral)